# Germline *AGO2* mutations impair RNA interference and human neurological development

Davor Lessel (ID) et al.[#]

ARGONAUTE-2 and associated miRNAs form the RNA-induced silencing complex (RISC), which targets mRNAs for translational silencing and degradation as part of the RNA interference pathway. Despite the essential nature of this process for cellular function, there is little information on the role of RISC components in human development and organ function. We identify 13 heterozygous mutations in *AGO2* in 21 patients affected by disturbances in neurological development. Each of the identified single amino acid mutations result in impaired shRNA-mediated silencing. We observe either impaired RISC formation or increased binding of AGO2 to mRNA targets as mutation specific functional consequences. The latter is supported by decreased phosphorylation of a C-terminal serine cluster involved in mRNA target release, increased formation of dendritic P-bodies in neurons and global transcriptome alterations in patient-derived primary fibroblasts. Our data emphasize the importance of gene expression regulation through the dynamic AGO2-RNA association for human neuronal development.

[#]A list of authors and their affiliations appears at the end of the paper.

RNA interference is a major mechanism for post-transcriptional regulation of gene expression. Precursors of microRNAs (miRNAs) are transcribed, processed into mature miRNAs and loaded onto Argonaute (AGO1-4) proteins to form the RNA-induced silencing complex (RISC)[1]. Each miRNA may recognize a set of target mRNAs by base pairing[2], which leads to translational silencing and mRNA degradation in cytoplasmic structures referred to as processing (P-) bodies[3–5]. Biallelic loss of *Ago2* leads to early embryonic lethality in mice exhibiting various development defects including anomalies of the central nervous system[6]. Therefore, accurate regulation of gene expression by the RNA interference pathway seems to be of utmost importance for proper development and maintenance of complex neural circuits[7,8]. However, so far no genetic alterations in the gene encoding for AGO2 have been described which are associated with any human pathology. Thus, it still remains largely elusive how RNA interference, and especially domains of AGO2 or local sequences down to single amino acid residues, regulate human organismal development and function.

Here, we demonstrate that germline *AGO2* mutations affect human neurological development and provide molecular insight into how AGO2 dysfunction causes a human Mendelian disorder.

## Results

**Identification of patients bearing germline *AGO2* mutations.** During trio whole-exome sequencing of a cohort of 50 children affected by developmental disturbances and neurological manifestations of unknown etiology[9], we identified a patient bearing a de novo missense mutation p.L192P in *AGO2* (NM_012154.5). Based on the ExAC and gnomAD sequencing data, *AGO2* is one of the most missense-intolerant genes in the human genome, ranked 15 and 30 respectively. Its Z scores of 7.696 (ExAC dataset) and 6.058 (gnomAD dataset) are far higher than the average Z score for genes involved in developmental

disorders[10–12]. Only two *AGO2* non-synonymous alterations (p.G88V and p.E186K) with an allele frequency of more than 0.0001 are deposited in ExAC and gnomAD datasets. Moreover, the p.L192P variant was not present in publicly available datasets (dbSNP, ExAC and gnomAD), had a high in silico pathogenic prediction score (CADD_Phred of 27.1) and changed a highly conserved residue (Supplementary Fig. 1). Furthermore, a previous study aiming to identify de novo variants in 20 individuals with sporadic non-syndromic intellectual disability identified a de novo p.L190P in *AGO1*[13]. The leucine at position 192 in *AGO2* corresponds to the leucine at position 190 in *AGO1* (Supplementary Fig. 1). In addition, five individuals affected by neurodevelopmental disturbances bearing a deletion that encompasses both *AGO1* and *AGO3* have been documented previously[14]. These findings motivated us to search for further cases carrying heterozygous *AGO2* variants utilizing both the internet-based GeneMatcher tool[15], and direct contact to our network of collaborators.

This approach led to identification of altogether 21 patients, including our index patient, affected by mild to severe global neurodevelopmental delay, who were discovered either by whole-exome sequencing or microarray-based comparative genomic hybridization. Eleven missense mutations, one in-frame deletion (Fig. 1a) and a 235.3-kb deletion involving the first three exons occurred de novo (Fig. 1b). In addition, one of the missense mutations (p.T357M) was transmitted from a similarly affected mother. Five mutations (p.L192P, p.G201V, p.T357M, p.M364T, p.C751Y) were recurrent. None of the mutations were present in publicly available datasets, and similarly to p.L192P, all missense mutations display high in silico pathogenic prediction scores (mean CADD_Phred of 29.1) and change highly conserved residues (Supplementary Data 1; Supplementary Fig. 1). Thus, the genetic data, especially the recurrence of mutations, already provide strong evidence for their pathogenicity.

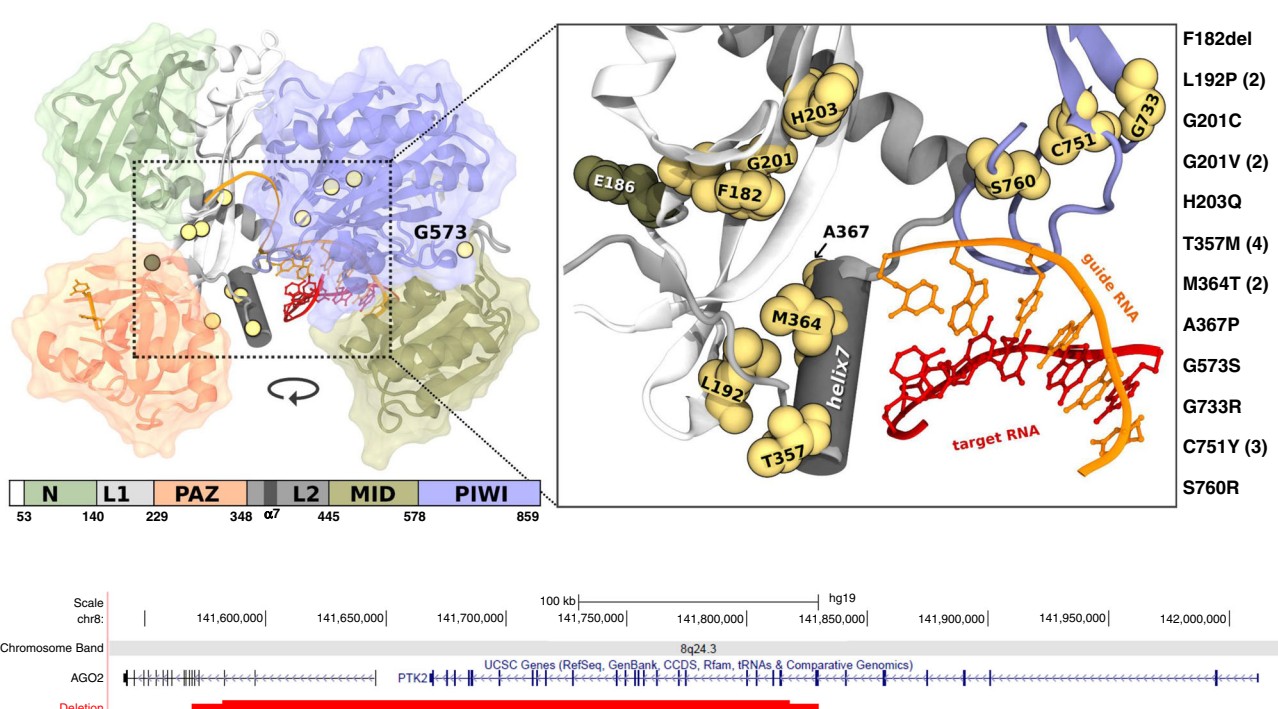

**Fig. 1 Location of identified *AGO2* germline mutations. a** Domain structure of AGO2 and position of the single amino acid mutations using the structure of human AGO2 in complex with a miRNA and a target RNA[22]. Guide and target RNA are depicted in orange and red, respectively. The recurring mutations are designated in brackets. **b** Genomic region, chr8.hg19:g.(141,582,269-141,817,600)del, of the 235.3-kb deletion identified in case 21, involving the first three *AGO2* exons and the last 23 *PTK2* exons.

**Table 1 Summary of clinical findings in individuals bearing *AGO2* mutations.**

| | Amount | Percentage |
|---|---|---|
| Neurological signs | | |
| Intellectual disability | 21/21 | 100 % |
| Motor developmental delay | 21/21 | 100% |
| Impaired speech development | 21/21 | 100% |
| Impaired receptive language | 13/13 | 100% |
| Muscular hypotonia | 12/21 | 57% |
| Autistic features | 9/16 | 56% |
| Cerebral MRI abnormalities | 9/16 | 56% |
| Gait abnormalities | 10/18 | 55% |
| Attention deficit hyperactivity disorder | 8/15 | 53% |
| Seizures | 8/18 | 44% |
| Strabism | 7/21 | 33% |
| Visual impairment | 6/21 | 29% |
| Abnormal respiration | 5/19 | 26% |
| Agressive behavior | 4/17 | 24% |
| Myopia/Hyperopia | 4/21 | 19% |
| Craniofacial abnormalities | | |
| Epicanthic folds | 11/21 | 52% |
| Thin upper lip | 11/21 | 52% |
| Dental anomalies | 9/19 | 47% |
| Frontal bossing | 9/21 | 43% |
| Open mouth appearance | 9/21 | 43% |
| Deep set eyes | 9/21 | 43% |
| Upslanting palpebral fissures | 6/21 | 29% |
| Congenital anomalies of the skull | 6/21 | 29% |
| Helix anomalies | 5/21 | 24% |
| Broad nasal bridge | 3/21 | 14% |
| Other findings | | |
| Neonatal feeding difficulties | 12/19 | 63% |
| Skeletal anomalies | 9/19 | 47% |
| Gastroesophageal reflux | 7/19 | 37% |
| Heart anomalies | 6/18 | 33% |

The 21 individuals show overlapping phenotypes, summarized in Table 1, and in more detail in Supplementary Note 1 and Supplementary Fig. 2. All individuals displayed intellectual disability, albeit of variable degree, as well as delayed motor development, impaired speech and receptive language development (13/13). Twelve patients had hypotonia (57%) and ten had gait abnormalities (10/18, 55%). Nine patients (9/16, 56%) showed features of autism spectrum disorder, including stereotypic and hand-flapping behavior, eight patients (8/15, 53%) showed features consistent with attention deficit hyperactivity disorder, and four (4/17, 24%) developed aggressive behavior, predominantly upon entering puberty. Eight patients developed seizures (8/18, 44%). Brain anomalies on MRI, mainly affecting the corpus callosum, were observed in nine patients (9/16, 56%). Vision problems included visual impairment in six (29%), strabismus in seven (33%) and myopia or hyperopia in four (19%) patients. Various breathing abnormalities were observed in five (5/19, 26%) and included central apnea in the postnatal period (observed in both cases carrying the p.L192P mutation), sleep apnea and hypopnea. Craniofacial abnormalities included epicanthic folds (57%), thin upper lip (52%), frontal bossing (43%), open mouth appearance (43%), deep-set eyes (43%), upslanting palpebral fissures (29%), congenital anomalies of the skull (29%) including plagiocephaly (5/21) and scaphocephaly (1/21), various ear helix anomalies (24%) and broad nasal bridge (14%). Dental anomalies were observed in nine (9/19, 47%). Twelve had neonatal feeding difficulties (12/19, 63%). Skeletal anomalies, not including congenital anomalies of the skull, were observed in nine patients (9/19, 47%). Notably, all three

individuals carrying the p.C751Y mutation had bilateral clinodactyly of the 5th finger. Seven individuals experienced gastroesophageal reflux (7/19, 37%). Heart anomalies were observed in six patients (6/18, 33%), whereas three of them had patent foramen ovale.

**Spatial clustering of residues affected by germline *AGO2* mutations**. AGO2 consists of N-terminal (N), Piwi/Argonaute/Zwille (PAZ), middle (MID) and PIWI (P-element induced wimpy testis) domains which are connected by linker regions L1 and L2 (Fig. 1a). The N domain is indispensable for RNA unwinding during RISC formation[16]. The PAZ domain binds to the 3′ end of the guide RNA and is involved in RISC activation[17,18]. The MID domain provides a binding pocket for the 5′ end of the guide RNAs[19]. The PIWI domain harbors a conserved catalytic core that cleaves the passenger strand, mediates protein-protein interactions needed for the enrollment of GW182, and regulates the interaction between RNA and the MID domain[20,21]. Binding determinants for miRNA (in the AGO2/miRNA RISC) and the structural changes occurring upon engagement of mRNA targets have been identified through structural analyses[22,23]. An α-helical segment in the L2 linker (helix-7, residues 359-369) is responsible for the proper positioning of the guide RNA during scanning of target mRNAs; here, residue I365 intercalates between guide RNA bases and induces a conformational change which is required for rapid target recognition[24]. We identified three mutations in helix-7: p.T357M (four patients), p.M364T (two patients) and p.A367P. All three residues face towards the L1 linker and are in close spatial proximity to L192 in L1, mutated in two patients (p.L192P). Upon target mRNA recognition, helix-7 and the PAZ domain move relative to the rest of the protein by 4 Å, to accommodate the target mRNA strand[22]. The hinge for this movement is partially localized at the base of L1, where we identified p.F182del, p.G201C, p.G201V (two patients) and p.H203Q; this is a region of the protein which has previously been implicated in the unwinding of RNA duplexes[16]. Finally, mutations p.G733R, p.C751Y (three patients, including monozygotic twins) and p.S760R affect residues which are in direct contact to each other, and located at the base of a loop in the PIWI domain which binds to the minor groove in the guide/target duplex[22]. Thus, we identify three spatial clusters of mutations: at the helix-7/L1 interface, at the hinge region of L1, and in a loop in PIWI which recognizes the guide-target duplex (Supplementary Fig. 3). Only p.G573S affects a residue outside of these clusters at the C-terminal end of the MID domain (Fig. 1a). In addition, the de novo (as determined by qPCR analysis) long-range deletion affecting the first three *AGO2* exons (Fig. 1b) is likely to result in haploinsufficiency (Supplementary Fig. 4), similar to deletions affecting both *AGO1* and *AGO3*[14].

***AGO2* germline mutations impair shRNA-mediated silencing**. To analyze how single amino acid mutations affect AGO2 function, we deleted *AGO2* in human HEK293T cells using *CRISPR/Cas9*-technology. Complete loss of AGO2 was confirmed by Western blotting (Supplementary Fig. 5). We used three shRNAs which had been established before in the lab and shown to be effective in reducing expression of Shank3, δ-catenin and DDX1, respectively. In line with the previously established essential role of AGO2 in shRNA-mediated gene silencing[25], strongly reduced silencing activity was observed for these shRNAs in AGO2 deficient cells, as they did not silence coexpressed Shank3 (Fig. 2a), DDX1 (Fig. 2b) and δ-catenin (Fig. 2c and Supplementary Fig. 6) mRNAs. Re-expression of WT-AGO2 efficiently rescued this phenotype, with residual expression of target proteins Shank3 of about 20%, and DDX1 and δ-catenin of <2% compared to

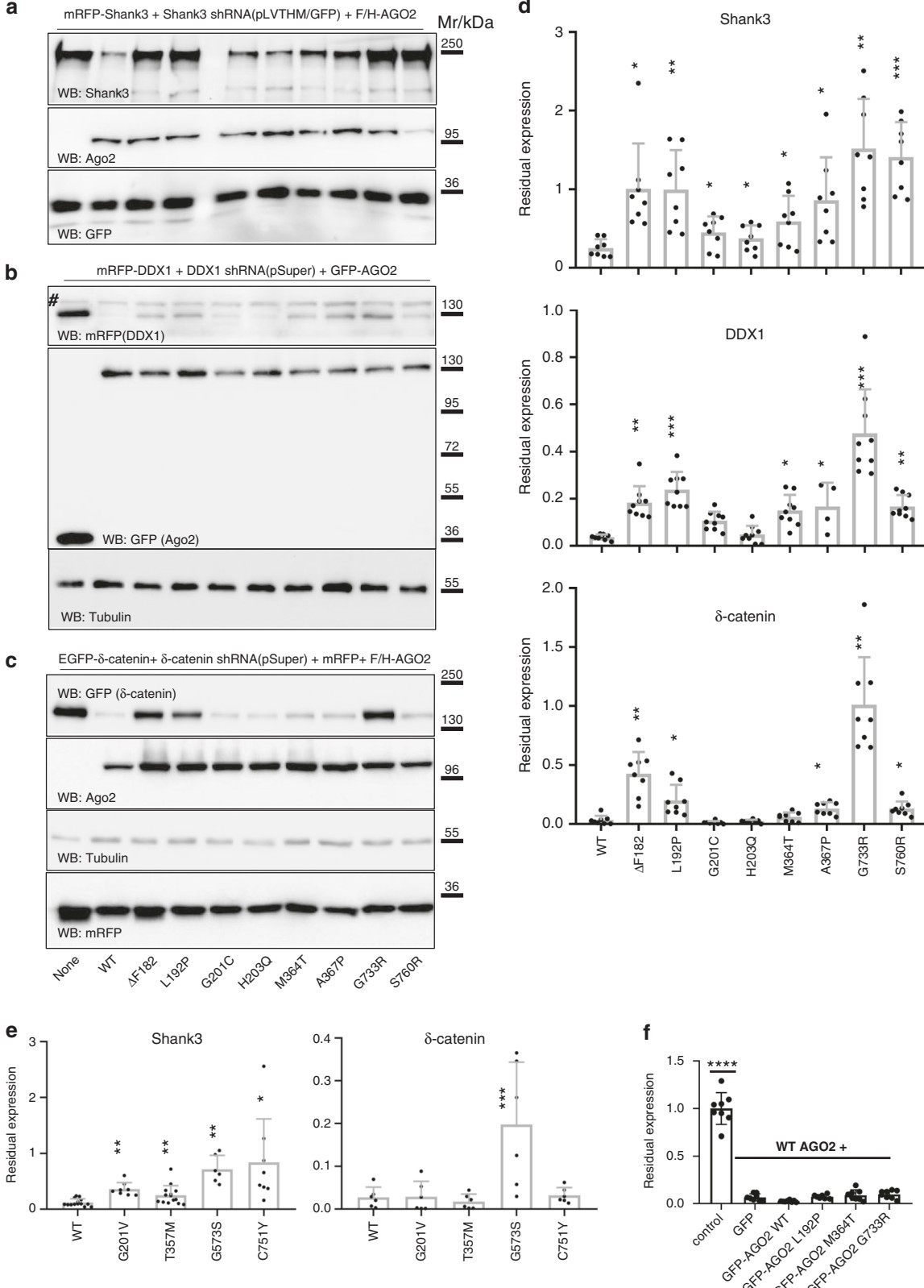

AGO2-deficient cells. All mutants were significantly less efficient than WT AGO2 in Shank3 silencing, whereas only some of the mutations showed significant effects for DDX1 and δ-catenin (Fig. 2a–e), suggesting target mRNA specific effects of some mutations. We further used this experimental system to investigate whether *AGO2* variants may exert a dominant-negative effect on the function of the WT AGO2 protein. Notably, we observed that knockdown of DDX1 expression by the DDX1 shRNA in the presence of F/H-tagged WT AGO2 was not affected by coexpression of selected mutant GFP-tagged AGO2 (p.L192P; p. M364T or p.G733R; see Fig. 2f). Thus, these data suggest that, at least the analyzed mutants, result in a loss-of-function.

**Fig. 2 Germline *AGO2* mutations impair shRNA-mediated silencing. a–c** AGO2-deficient HEK293T cells were transfected with mRFP-Shank3 (**a**), mRFP-DDX1 (**b**), or GFP-tagged δ-catenin (**c**) along with shRNA constructs targeting Shank3 (**a**; in pLVTHM vector coexpressing GFP), DDX1 (**b**; in pSuper) or δ-catenin (**c**; also in pSuper), and Flag/HA-tagged (Shank3, δ-catenin) or GFP-tagged (for DDX1) AGO2 variants as indicated. Efficient transfection (>70 % of cells) was verified by fluorescence microscopy using appropriate fluorophores (GFP in (**a**) and (**b**); RFP in (**c**). Cells were lysed and lysates were analysed by Western Blotting using the antibodies indicated. In each case, each experiment was repeated eight times with similar results. **d** Quantification of the representative immunoblots shown in (**a–c**). The expression levels are relative to those in control cells without an AGO2 construct (first lane in each immunoblot). **e** Quantification of a second set of AGO2 mutants. For (**d**, **e**), data are means + SEM. Data for mutant and control conditions were compared to wt. *, **, ***$p < 0.05, 0.01, 0.001$, respectively; $n = 8$ biologically independent experiments in (**d**); one-way-ANOVA, followed by Holm-Sidak's multiple comparisons test; $n = 6$–14 biologically independent experiments in e for Shank3; mixed-effects analysis, followed by Holm-Sidak' multiple comparisons test; $n = 6$ biologically independent experiments in (**e**) for δ-catenin; one-way ANOVA, followed by Dunnett's multiple comparisons test. **f** Knockdown of DDX1 with DDX1 shRNA was performed as in (**b**), but using F/H-tagged AGO2 in combination with GFP, or in combination with GFP-tagged AGO2 variants. Note that the GFP-tagged AGO2 mutants do not interfere with the knockdown capacity of F/H-AGO2 WT. $n = 8$ biologically independent experiments; data are means +/− SD; ****$p < 0.0001$; one-way ANOVA, followed by Dunnett's multiple comparisons test. Source data are provided as a Source Data file.

As the abovementioned experiments were performed by overexpression of AGO2 variants, we next carefully calibrated our experimental system over a wide range of WT AGO2 concentrations. In brief, we transfected different amounts of AGO2 expression vectors into the AGO2 deficient cells, while keeping amounts of cotransfected shRNA (for δ-catenin) and target vector (GFP-δ-catenin) constant. AGO2 expression relative to native HEK293T cells ranged from 54 % for WT AGO2, when the lowest concentrations of DNA was used (0.1 μg), to an about 20-fold overexpression when 1 μg was transfected into each well of a 12 well plate (Supplementary Fig. 6). Interestingly, the silencing efficiency of WT AGO2 was maintained almost unaltered over this wide range of concentrations. Even at expression levels which are lower than those in unmodified HEK293T cells, efficient downregulation of δ-catenin was observed. Importantly, both at low and at high concentrations of AGO2, differences between WT and mutant forms of AGO2 were maintained, such that even at more than 10-fold over-expression, the p.F182del, p.L192P, and p.G733R variants did not silence as efficiently as AGO2 WT at its lowest concentration (Supplementary Fig. 6).

We conclude from these titration experiments that strong differences between WT and mutant forms of AGO2 are observed at physiological levels of the protein (i.e., at levels found in the non-modified HEK293T cell line).

**RISC formation is differentially affected by germline *AGO2* mutations.** Having established a functional deficit for all single amino acid mutations, we used a spectrum of assays to determine which step of the miRNA pathway might be affected. We tested the ability of AGO2 variants to interact with DICER, the protein responsible for pre-miRNA processing and loading of the mature miRNA onto AGO2 (Supplementary Fig. 7), and with the proteins of the TNRC6 family that associate with AGO proteins during miRNA-guided gene silencing (Supplementary Fig. 8). We further assessed the capacity to bind to an endogenous miRNA (miR19-b) in HEK293T cells (Supplementary Fig. 8), the nuclease or slicing function of the AGO2 variants in in vitro cleavage assays using a radiolabeled substrate[26] (Supplementary Fig. 9), and the ability to silence a luciferase-based reporter mRNA in Hela cells (Supplementary Fig. 10). Finally, correct targeting of AGO2 to P-bodies[3,27] was investigated in U2OS cells (Supplementary Fig. 11). In these assays, most mutants performed similar to WT-AGO2, with the exception of p.G733R, which appeared to be non-functional in almost every aspect tested. Thus, mutations in AGO2 have two functional consequences: first, in the majority of the mutants, basic aspects of RISC formation and AGO2 function are not affected, and second the p.G733R mutant

exhibits a loss-of-function in almost every assay, likely similar to the anticipated effect of the here identified 235.3-kb deletion.

**Reduced phosphorylation and altered mRNA target release of germline *AGO2* mutations.** AGO2 function is regulated by protein kinases whereby phosphorylation at Ser387 by AKT3 favors translational repression of targets over degradation[28]. Importantly, phosphorylation of a C-terminal cluster (S824–S834) by casein kinase α1 (CSNK1A1) is associated with the final step of AGO2-mediated mRNA repression, which enhances the release of the target mRNA from the RISC complex[29,30]. We measured phosphorylation of Ser387 and the C-terminal cluster upon immunoprecipitation of AGO2 from HEK293T cells, followed by quantitative mass spectrometry. Phosphorylation of S387 was not altered, whereas phosphorylation of the C-terminal cluster was strongly reduced in all mutants tested, except for p.H203Q (Fig. 3a). By performing qRT-PCR on RNA samples isolated from AGO2 immunoprecipitates, we further observed that all mutants exhibiting reduced phosphorylation of this cluster also showed an increased association with a set of known target mRNAs, with the exception of p.G733R, which did not bind any mRNAs further supporting a complete loss of function of this mutant (Fig. 3b). These data further corroborate the link between phosphorylation of the S824-S834 cluster and the mRNA release from AGO2[29,30], and indicate a slower release of the majority of the identified AGO2 mutants from target mRNAs. Thus, we suggest a model in which a reduced phosphorylation of the C-terminal serine cluster of most of the disease-causing mutants coincides with reduced target release and thus an extended dwelling time of the AGO2 mutants on their targets.

**Molecular dynamics simulations suggest an effect of germline *AGO2* mutations on AGO2 unwinding function.** To further clarify the effect of *AGO2* mutations, apart from the clear loss-of-function mutation p.G733R, we performed non-biased molecular dynamics (MD) simulations. Our goal was to gain further insight into the effect of the above-mentioned mutations in apo-AGO2 and AGO2-RNA complexes. As a positive control, we included p.F181A which was shown before in an alanine scanning muta-genesis to reduce the unwinding of both siRNA and miRNA duplexes[16]. As a negative control, we included p.E186K, one of the two common non-synonymous *AGO2* variants found in public repositories. *AGO2* mutations were simulated in five states corresponding to different complexes along RISC formation: (i) apo-AGO2; two AGO2 complexes with guide RNA (core-RISC), namely intercalating (ii; *int* core-RISC) and non-intercalating (iii; *non- int* core-RISC); and two AGO2 complexes with guide-target duplexes (holo-RISC) with fully matched seed region (iv; g2-7

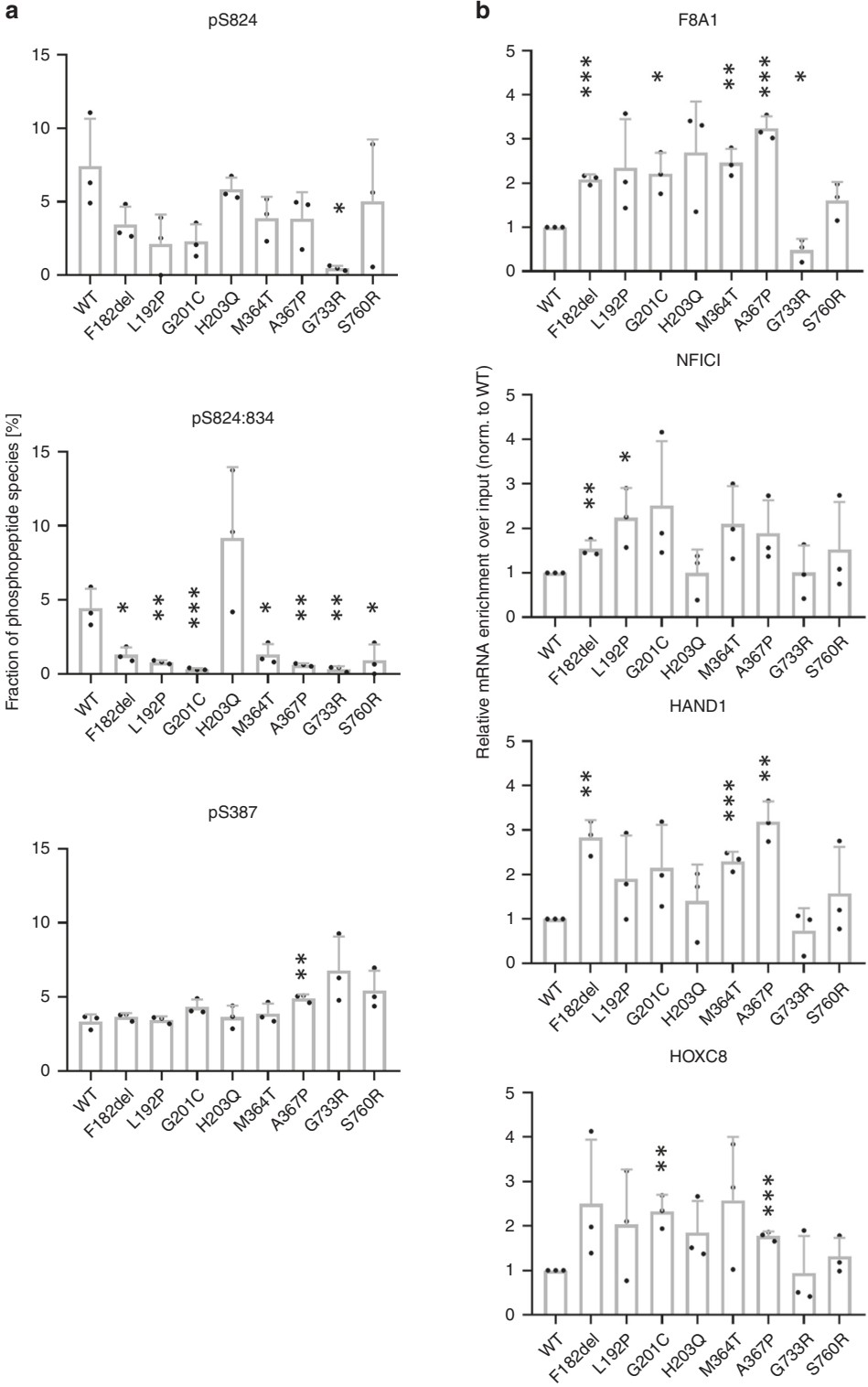

**Fig. 3 Reduced target dissociation of *AGO2* germline mutants. a** 293 cells were transfected with F/H-tagged AGO2. After immunoprecipitation, phosphorylation of S824 alone, of the S824–S834 cluster and of S387 was measured by a targeted quantitative mass spectrometry approach (Selected reaction monitoring with isotopically labeled spike-in peptides). The *y*-axis represents the percentage of individual phosphorylated peptide species assuming the sum of singly, multiply and non-phosphorylated peptides to be 100%. Significance was assessed by two-sided Student's *t* test in relation to WT. $n = 3$ biologically independent experiments. Data are presented as mean $+$ SD. *, **, ***$p < 0.05$, 0.01, 0.001, respectively. **b** RNA was isolated from F/H–AGO2 immunoprecipitates and analyzed by qRT-PCR using primers for the genes indicated. The significance was assessed by two-sided Student's *t* test in relation to WT. $n = 3$ biologically independent experiments. Data are presented as mean $+$ SD. *, **, ***$p < 0.05$, 0.01, 0.001, respectively. Source data are provided as a Source Data file.

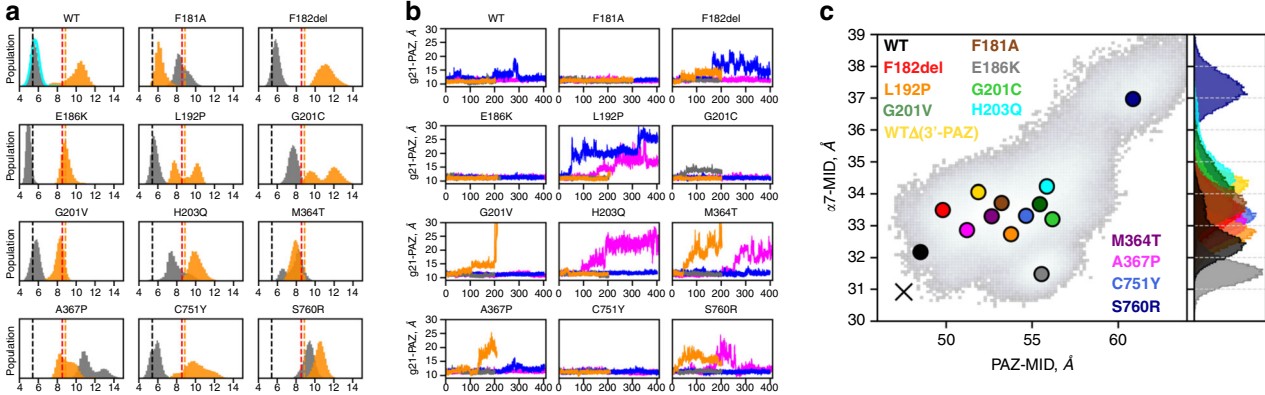

**Fig. 4 Molecular dynamic simulation of effects of *AGO2* mutations. a** Distance between the Cα atom of I365 and the glycosidic N atom of the g7 residue of the guide (I365α-g7(N)). Populations of all core-RISC trajectories were normalized to the same bin number (60). Black and orange dashed vertical lines denote the reference X-ray structures of the *int* and *nonint* core-RISC states (4OLA and 4W5N, respectively). The red dashed line denotes the X-ray structure of the helix-7 mutant (5WEA). **b** Distance between the c.o.m. of the 3′-end of the guide and the c.o.m. of the PAZ domain (g21-PAZ) along the trajectories of all RNA-bound states. Note that the relatively short length of the individual MD trajectories could affect the results. Color code of the states in both a and b panels: apo-Ago2— green, *int* core-RISC—gray, *nonint* core-RISC—orange, g2-8 holo-RISC—magenta and g2-7 holo-RISC—blue. **c**d motion of the helix7 and the PAZ domain along the open-closed mode (left panel, gray scale histogram). The histograms are calculated by concatenating the last 100 ns of non-biased trajectory of each variant with mismatched RNA duplex. Colored circles depict the maximum population density of each trajectory. Black cross denotes the maximum population density of the WT AGO2 with guide RNA. Right panel: population histograms on α7-MID corresponding to the maxima on the left panel. Equivalent analysis of the variants in in complex with a fully matched seed duplex is shown on Fig. S16.

holo RISC) or with a mismatch at position 8 of the guide (v; g2-8, holo RISC). The underlying structures and pdb codes are listed in Supplementary Fig. 12. This large number of simulations (12 *AGO2* variants in five different states) was chosen as an initial screening approach to identify possible alterations in mutant forms of AGO2; however, this large number also precluded detailed quantitative measurements. Among the simulated complexes, the mutations appeared not to uniformly affect the population distributions on this conformational mode. Principal component analysis of apo-AGO2 trajectories suggested that only two mutations, p.L192P and p.F182del, affected global protein dynamics. The predominant conformational mode of these two variants deviated from the open-closed mode, previously described as the most pronounced conformational mode of AGO proteins[31] (Supplementary Fig. 13). Further, analyses of the non-biased MD trajectories of RNA-bound complexes suggested two effects: (i) a compromised interaction in the *int* core-RISC state between helix-7 and g7 which was observed for the mutations p.G201C, p.H203Q, p.M364T, p.A367P and p.S760R, similar to the p.F181A positive control (Fig. 4a); and (ii) a loss of anchoring of the 3′-end of the guide (g21) at the PAZ domain, observed for all patient-derived mutations apart from the p.C751Y in at least one simulated state (Fig. 4b). The first effect (on the helix-7-g7 interaction) is somewhat reminiscent of a previously reported mutation in helix-7, which was shown to affect rapid target recognition by RISC[24]. The I365α-g7 distance derived from the 3D structure of this mutant was therefore included here in Fig. 4a for comparison.

Based on these findings, we next addressed the guide-target duplex unwinding function of AGO2. First, we performed 360ns-long 1D metadynamics (MetD) simulations of WT and p.L192P in the duplex-bound complex, in which we enhanced sampling of I365 intercalation between g6 and g7 using I365δ-g(6,7) distance as a collective variable (see Supplementary Fig. 14a for the definition of collective variables). Here, we used the energy potentials deposited during the MetD simulations to induce intercalation-mediated partial duplex unwinding, as we could not calculate the free energy profiles of intercalation. Therefore, the MetD here provides only a qualitative picture of the possible unwinding mechanism, allowing us to speculate which aspects of

this process are altered by the patient-derived mutations. As a simplified metric of the unwinding progress, we used mean guide-target duplex width (<C1′-C1′>) (Supplementary Fig. 14b). During unwinding, helix-7 appears to move towards the MID domain and 'squeeze' the RNA duplex at base pairs g4-g7 (Supplementary Fig. 14a, 14c), thereby pushing the two RNA strands apart (Supplementary video 1). In WT-AGO2 the highest duplex width reached is slightly larger than in p.L192P (~13 Å vs. ~11.5 Å, respectively), as helix-7 in WT-AGO2 shifted closer to the MID domain (Supplementary Figs. 14b and 15). However, the nature of this analysis allowed only for the qualitative comparison between WT and p.L192P. The movements of the PAZ domain and helix- 7 towards the opposing AGO2 lobe (MID and PIWI domains) are concerted (Fig. 4c, gray histogram in the background), corroborating previous suggestions[22,32].

Population distributions of the non-biased MD trajectories of variants bound to a mismatched duplex (g2-7 holo-RISC) suggest that compared to WT-AGO2, helix-7 shifted further away from the MID domain—an effect that seems to be common among all AGO2 mutants (Fig. 4c; see also Supplemental Fig. 16). Furthermore, the helix-7-MID distance appeared larger also for the p.F181A positive control (~1.6 Å larger than WT), but not for the common non-synonymous *AGO2* variant p.E186K. Moreover, a similar effect is also observed when we manually removed the guide 3′-end from the PAZ domain of the duplex-bound state (WTΔ(3′-PAZ) in Fig. 4c). This provides a further link between loss of anchoring of the guide 3′-end at the PAZ domain, observed for almost all here identified mutations, and AGO2-mediated guide-target duplex unwinding[33]. Taken together, based on our simulations, we hypothesize that the mutations identified in patients, apart from p.G733R, lead to reduced target release due to impaired AGO2 unwinding function. However, one should keep in mind that our data are mostly qualitative, and further, more quantitative simulations will be needed to completely dissect how the mutations in *AGO2* affect unwinding.

**AGO2 germline mutations lead to aberrant density of dendritic P-bodies.** GFP-tagged WT-AGO2 and a representative set of

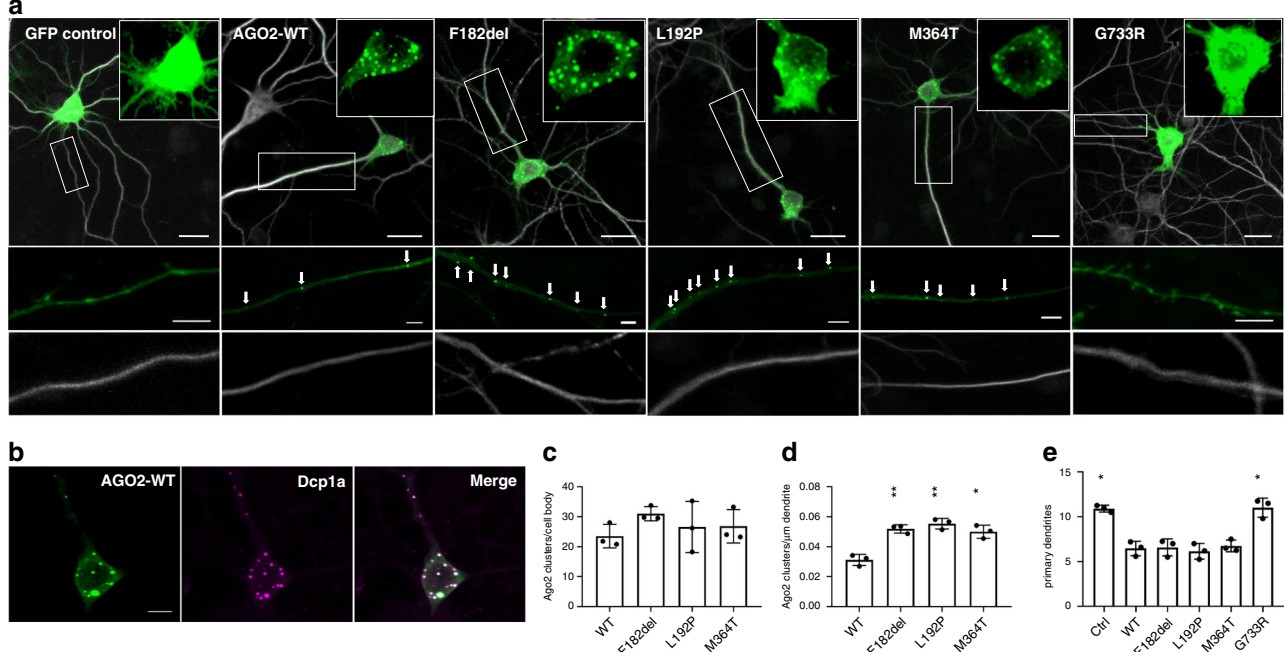

**Fig. 5 AGO2 germline mutations lead to an increased number of dendritic P-bodies. a** Primary cultured murine hippocampal neurons were transfected with GFP control or GFP-AGO2 WT and variants and Tomato-red. Staining for the dendritic marker MAP2 is shown in gray; inserts display the GFP signal in cell bodies. Boxed areas are magnified below for GFP- and MAP2 signals. Arrows indicate dendritic GFP-AGO2 clusters. Similar results were obtained in three independent biological experiments; results are quantified in (**c**, **d**, **e**). Scale bars: 20 μm in overview pictures; 5 μm in inserts. **b** Neurons expressing GFP-AGO2 were co-stained for Dcp1a. Similar costaining results were obtained in two biologically independent experiments for wt and F182del, L192P and M364T mutants, with 30 cells analyzed per experimental condition Scale bar: 5 μm. **c–e** Quantification of GFP-AGO2 clusters in cell bodies (**c**), GFP-AGO2 clusters in dendrites (**d**), and primary dendrites per cell (**e**). Data for mutant and control conditions were compared to wt. *, **$p < 0.05$, 0.01, respectively; Brown–Forsythe and Welch one-way-ANOVA, followed by Dunnett's T3 multiple comparisons test; $n = 3$; mean ± SD). Source data are provided as a Source Data file.

identified mutations were expressed in primary cultured rat hippocampal neurons. Cells were stained for the dendrite marker MAP2 (Fig. 5a), synaptic marker Shank3 (Supplementary Fig. 17), and the P-body marker Dcp1a (Fig. 5b). WT-AGO2 formed multiple clearly isolated punctae in cell bodies and throughout dendrites, consistent with previous observations[34–36]. This pattern was maintained for p.F182del, p.L192P and p. M364T mutants but not for p.G733R, which appeared entirely diffuse similar to the GFP-control protein (Fig. 5a). In agreement with the data from non-neuronal cells (Supplementary Fig. 11), all GFP-AGO2 puncta were identified as P-bodies by co-staining for Dcp1a (shown for WT in Fig. 5b). Importantly, we found that whereas the number of AGO2-containing granules in neuronal cell bodies was not affected by the mutations (Fig. 5c), the density of dendritic P-bodies was almost doubled upon expression of mutants p.F182del, p.L192P and p.M364T (Fig. 5d). We conclude that the reduced phosphorylation at the C-terminal serine cluster, and the delayed dissociation from mRNA targets leads to an increased presence of AGO2 at dendritic P-bodies. This indicates that mutations in AGO2 lead to impairment of local translation that may result in altered plasticity in response to synaptic activity[34–36]. Morphological analyses showed that neurons expressing WT, p.F182del, p.L192P and p.M364T exhibited a reduced number of dendrites compared to GFP-control and p. G733R (Fig. 5a, e). It has been shown previously that expression of WT-AGO2 is capable of reducing dendritic complexity[37], and again only the p.G733R failed to do so. The number of Shank3 positive clusters in dendrites was similar for all variants, indicating that synaptogenesis is not affected by over-expression of WT-AGO2 and the AGO2 mutants (Supplementary Fig. 17).

**Global transcriptome alteration in primary fibroblasts of AGO2 patients.** Given the major role of AGO2 in post-transcriptional regulation of gene expression, we next assessed global changes in the transcriptome in patient-derived primary dermal fibroblasts. RNA sequencing was performed of primary dermal fibroblasts obtained from two patients bearing the p. L192P (cases 2 and 3) and one patient with the p.A367P mutation (case 14). We compared the expression patterns of the protein-coding transcripts to fibroblasts from age-matched individuals, who were either unaffected or bear a causative mutation unrelated to AGO2. More than 770 genes were differentially expressed (DE) in case 2 compared to his three age-matched controls, whereas more than 1500 genes were differentially expressed in both cases 3 and 14, each of which was compared to a single age-matched control. 485 DE genes overlapped between cases 2 and 3, who bear the identical de novo p.L192P mutation (Supplementary Data 2 and 3). All three AGO2-patients shared 164 commonly DE genes (Fig. 6a, b, Supplementary Data 4), suggesting common but also mutation-specific effects. Gene ontology enrichment analysis of the commonly DE genes revealed enrichment for terms related to mitosis and cell cycle regulation (Supplementary Data 5), which is in line with the previously described function of AGO2 in regulating accurate chromosome segregation and cell cycle progression[38,39].

**Discussion**
In this study, we present clinical and molecular findings in 21 individuals bearing heterozygous AGO2 mutations, accompanied by in-depth functional characterization of identified missense mutations. All individuals exhibited intellectual disability and

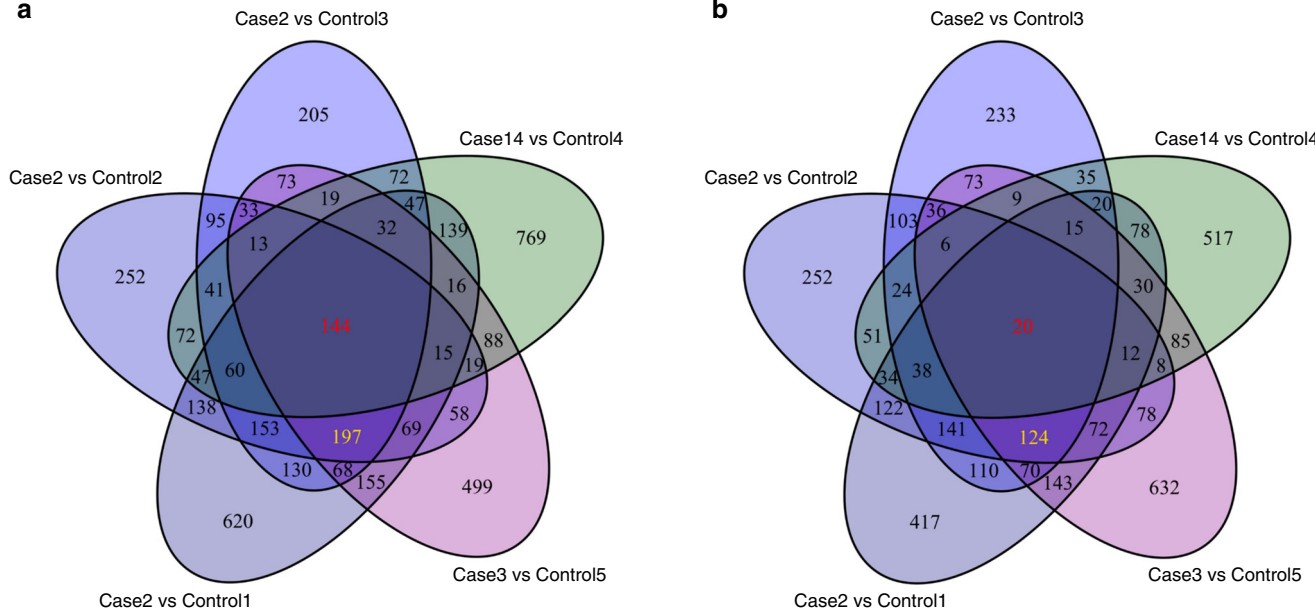

**Fig. 6 Global transcriptome alteration in primary fibroblasts of *AGO2* patients. a**, **b** Venn diagram of upregulated (**a**) and downregulated (**b**) transcripts in fibroblasts isolated from *AGO2* patients (cases 2, 3, and 14) compared to five age-matched controls. The number of the upregulated (**a**) or downregulated (**b**) genes in all three cases are marked red, whereas deregulated genes in both cases bearing the p.L192P mutation are marked yellow.

developmental delay, including delayed motor development and impaired speech development. Moreover, more than half of the patients had neonatal feeding difficulties and hypotonia. The majority of the patients presented with behavioral abnormalities including features of autistic spectrum disorder and attention deficit hyperactivity disorder. Some individuals developed aggressive behavior upon entering puberty. Seizures were a further common finding, observed in almost half of the individuals described here. Although most displayed some dysmorphic features, we did not observe a recognizable facial pattern. An increased incidence of skeletal and heart anomalies was noted. Interestingly, five individuals presented with plagiocephaly, a feature which may be related to hypotonia in four individuals. Furthermore, some specific clinical signs and symptoms were observed exclusively in individuals bearing the identical mutation. These include bilateral clinodactyly of the 5th finger as well as misaligned or crowded teeth with large incisors in all three individuals carrying the p.C751Y, and central apnea in the postnatal period in both individuals bearing the p.L192P. We additionally observed some unique clinical signs and symptoms. Individual 3 presented with a differential diagnosis of Pallister–Killian syndrome, individual 14 had blue sclera, individual 15 developed hydronephrosis, and individual 20 presented with precocious puberty. Future clinical reports beyond the large number of the affected individuals identified in this study will both broaden genotype-phenotype correlations and likely identify further mutation-specific clinical features.

Our functional analyses show a clear reduction of activity of the here identified *AGO2* mutations in an shRNA-based silencing assay. This corroborated the in silico pathogenic predictions. In several cases we observed only subtle changes, which is consistent with the overall extreme intolerance of human *AGO2* to mutations. Thus, our data show that even minor deficits in AGO2 function are sufficient to elicit aberrant neurological development. Currently, our characterization of selected mutations supports the view that they lead to a loss-of-function rather than a dominant-negative effect (see Fig. 2f). This is consistent with the

fact that so far only a single homozygous, non-synonymous *AGO2* variant carrier (bearing the p.G88V) is documented in the gnomAD dataset.

Notably, one of the mutations identified here, p.G733R, stands out from the others as it fails in almost every functional assay. We assume that the substitution of the rather bulky and positively charged Arg for the small Gly will cause structural perturbations and local unfolding in the PIWI domain, thus leading to a complete loss of function. In this respect, we predict it to be functionally similar to the large chromosomal deletion observed in patient 21. The deletion encompasses part of the *AGO2* gene and is likely to lead to haploinsufficiency. It does not only lead to loss of the first 3 *AGO2* exons, but also the last 23 exons of *PTK2*. To our knowledge, no mutation in *PTK2* has so far been connected to any human disorder. Several individuals bearing heterozygous loss-of-function mutations in *PTK2* have been documented in the gnomAD dataset. Nonetheless, since no additional individuals with similar *AGO2* loss-of-function mutations, (either a gross deletion, nonsense or frameshift mutation), are known currently, we cannot deduce to which extent the additional *PTK2* deletion might be contributory. It is worth noting that the clinical presentation of this individual somewhat resembles children bearing a deletion that encompasses both *AGO1* and *AGO3*[14].

Currently, the pathomechanism of the p.G573S mutation remains somewhat unclear. It is conceivable that the change from glycine to serine, due to the difference in the physico-chemical properties of these residues, may cause some local structural changes and alter the interface between PIWI and MID domains.

Importantly, our analyses add physiological and pathological significance to recent advancements in structural and mechanistic understanding of ARGONAUTE proteins. The fact that most of the pathogenic variants in *AGO2* alter residues either in L1 or helix-7 of L2, corroborate the structural analyses showing that subtle movements at the interface between these two segments determine the kinetics of target recognition[22,24]. Furthermore, phosphorylation of the serine cluster at residues 824 to 834 has

recently been shown to coincide with the fast release of targets from RISC[29,30]. In the AGO2 mutants, we observe reduced phosphorylation of this cluster, which together with our MD simulations indicate that the unwinding of guide-target duplexes may be slowed down by mutations, due to reduced movement of helix-7 towards the MID domain of the protein.

The reduced phosphorylation and enhanced binding to target mRNAs of mutant AGO2 coincide with an enhanced appearance of AGO2-positive dendritic P-bodies in neurons, as observed here for p.F182del, p.L192P and p.M364T variants. This may contribute to the neuronal phenotype seen in patients, as dendritic P-bodies and the activity of AGO proteins in dendrites are believed to contribute to synaptic activity[34–36]. In particular, it is conceivable that changes in dendritic P-bodies will alter the complement of dendritic mRNAs which may be locally translated upon synaptic stimuli. Further work will be needed to determine how this affects neuronal function and morphology, as well as synaptic plasticity, learning and memory.

Our observation that patient-derived fibroblast cells exhibit global alterations of gene expression support the view that the global transcriptome changes due to altered function of AGO2 protein. Interestingly, we observed enrichment for commonly differentially expressed genes with broad roles in mitosis and cell cycle regulation, including cell division, mitotic nuclear division, sister chromatid cohesion, chromosome segregation, microtubule binding and regulation of cell cycle, to highlight only a few. Notably, the causal link of neurodevelopmental disorders and genes encoding for proteins regulating abovementioned processes have been previously established[40–45]. It would be interesting to delineate which of these biological processes are affected in patient primary fibroblasts. Moreover, given the different organ- and tissue-specific expression patterns, a further emerging question is whether similar transcriptome changes will be observed in iPSC-derived neurons from these patients or even in patient-derived cerebral organoids. In addition, further studies are needed to delineate changes in global miRNA expression and investigate if these pathogenic mutations result in different RNA and miRNA binding sites. These analyses will require extensive further work and will be the main aim of our future studies.

Taken together, our study demonstrates that mutations affecting a core component of the RNAi machinery are associated with altered human neurological development, supporting previous observations that development and function of the nervous system is particularly vulnerable to alterations in gene expression patterns and their regulation[7].

## Methods

**Research subjects**. Written informed consent for all subjects was obtained in accordance with protocols approved by the respective ethics committees of the institutions involved in this study (approval number by the Ethics Committee of the Hamburg Chamber of Physicians: PV 3802). The authors affirm that the research participants and their legal representatives, and in the case of minors the parents or legal representatives of the human research participants provided informed consent for publication of the images in Supplementary Fig. 2.

**Genetic analyses**. Some of the investigators presenting affected individuals in this study were connected through GeneMatcher, a web-based tool for researchers and clinicians working on identical genes[15]. Whole-exome sequencing (WES) or trio whole-exome sequencing (trio-WES) experiments, data annotation and interpretation were performed in nine different centers with slightly different procedures. Trio-WES in families of cases 4, 6, 7, 8, 12, 13, and 16 was performed at the Radboud University Medical Center in Nijmegen, the Netherlands[46]. Exome capture was performed with the Agilent SureSelect Human All Exon v5 enrichment kit (Agilent Technologies). Whole-exome sequencing was performed on the Illumina HiSeq platform (BGI, Copenhagen, Denmark) and the Illumina Nova-Seq6000 (NSW Health Pathology Randwick Genomics, Sydney, Australia). Data were analysed with BWA (read alignment,) and GATK (variant calling) software packages. Variants were annotated using an in-house developed pipeline.

Prioritization of variants was done by an in-house designed 'variant interface' and manual curation. Trio WES in families of cases 11, 15, 19, and 20 and quad-WES in a family of cases 17 and 18 were performed on exon targets isolated by capture using the SureSelect Human All Exon V4 (50 Mb) or the IDT xGen Exome Research Panel v1.0. Massively parallel (NextGen) sequencing was done on an Illumina system with 100 bp or greater paired-end reads. Reads were aligned to human genome build GRCh37/UCSC hg19 (for case 11 to GRCh38/UCSC hg38), and analyzed for sequence variants using custom-developed analysis tools[47]. The general assertion criteria for variant classification are publicly available on the GeneDx ClinVar submission page (http://www.ncbi.nlm.nih.gov/clinvar/submitters/26957/). Trio-WES in the family of case 2 was performed using a SureSelect Human All Exon 50 Mb V5 Kit (Agilent, Santa Clara, CA, USA), and sequencing was performed on a HiSeq2500 system (Illumina, San Diego, CA, USA). Reads were aligned to the human genome assembly hg19 (UCSC Genome Browser) with Burrows-Wheeler Aligner (BWA, v.0.5.87.5), and detection of genetic variation was performed using SAMtools (v0.1.18), PINDEL (v 0.2.4t), and ExomeDepth (v1.0.0)[48]. Trio-WES in family of case 14 was performed using SureSelect Human All Exon 50 Mb kit (Agilent Technologies, Santa Clara, CA) on a HiSeq2500 system (Illumina, San Diego, CA, USA). In-house developed scripts were applied to detect protein changes, affected splice sites and overlaps to known variations, with filtering against dbSNP build 138, the 1000 Genomes Project data build November 2014 and ExAC Browser (status from August 2019)[49]. DNA of case 9 and his unaffected father was sequenced SureSelect Clinical Research Exome V2 (Agilent Technologies, Santa Clara, CA) on a HiSeq2500 system (Illumina, San Diego, CA, USA). The raw data were analyzed using the Care4Rare analysis pipeline[50]. DNA of case 1 and his parents were extracted from peripheral blood using the Gentra Puregene Blood Kit (Gentra Systems Inc., Minneapolis, USA). Quality and quantity were checked with NanoDrop Spectrophotometer (Nano-Drop Technologies, Wilmington, USA). Genomic DNA was fragmented by sonication to generate fragments of 200-500 base pairs. Library preparation was done using Kapa DNA HTP Library Preparation Kit (KAPA Biosystems, 07138008001). Hybridization of the adapter-ligated DNA was performed to a biotin-labeled probe included in the Nimblegen SeqCap EZ Human Exome Kit (Roche, 06465692001). Libraries were sequenced using the Illumina Hiseq 2500 sequencing system and paired-end 101 bp reads were generated for analysis. Trio-WES in family of case 3 was performed using the SureSelect Human All Exon V4 (50 Mb) kit (Agilent, Santa Clara, CA, USA), and sequencing was performed on a HiSeq2500 system (Illumina, San Diego, CA, USA). Variants were identified using haplotype caller within GATK and Freebayes. The intersection of the two variant callers were annotated with SnpEff and loaded into a database using the GEMINI framework. Annotations included predicted functional effect (e.g., splice-site, nonsense, missense), protein position, known clinical associations (OMIM, CLINVAR), mouse phenotypes (MGI), conservation score (PhastCons, GERP), and effects protein function (PolyPhen), CADD scores, and population allele frequencies (Exome Variant Server and Exome Aggregation Consortium data). Trio-Wes in family of case 5 was performed on MGISEQ-2000 platform (BGI-Wuhan, Wuhan, China). DNA extraction as well as quality controls were performed using standard methods. Half a microgram of genomic DNA was randomly fragmented by Covaris. Using Agencourt AMPure XP-Medium kit fragments of 150-250 bp were selected. The fragments were then subjected to end-repair, 3′ adenylation and adaptors ligation. After PCR amplification and purification, hybridization using BGI Hybridization and Wash kits were used. After a second PCR and recovering step, the double stranded PCR products were heat-denatured and circularized by the splint oligo sequence. The single-strand circle DNA (ssCir DNA) were formatted as the final library. Library was qualified by Qubit ssDNA kit. The library was amplified to make DNA nanoball which have more than 300 copies of one molecular. The DNBs were loaded into the patterned nanoarray and pair-end 100 bases reads were generated in the way of sequenced by combinatorial Probe-Anchor Synthesis (cPAS). The raw data were then transferred to Limbus (Rostock, Germany) and bioinformatic secondary and tertiary analyses were performed using the standard algorithms of GATK. SNVs and CNVs were then presented in a web-based interface and evaluation was performed by experienced scientists. Identified candidate genes were then prioritized based on in-house scoring system.

Chromosomal Microarray Analysis in family 21 was performed using a 4×180 K whole-genome oligonucleotide microarray following the manufacturer's protocol (Agilent Technologies, Santa Clara, CA, USA). Results were interpreted with Cytogenomics software v3.0.1.1 (ADM2 method). A CNV was defined as at least three contiguous oligonucleotides with an abnormal mean log ratio (>0.25 or < −0.25). GRCh37/hg19 was used as the reference sequence.

**Plasmids**. An expression vector for GFP-tagged human AGO2 (in pEGFP-C1) was obtained from Phil Sharp (MIT) via Addgene (#21981[51]). The vector for Flag/HA (F/H) tagged human AGO2 has been described before[26]. Mutations were introduced by site-directed mutagenesis using the QuikChange II kit (Agilent; CA), using complementary oligonucleotides. Constructs were verified by Sanger sequencing. For expression in neurons, the entire cDNA fragments coding for GFP-AGO2 fusion proteins were subcloned into FUW vector which uses a ubiquitin promoter for driving expression. FUW was obtained via Addgene #14882 from D. Baltimore, Caltech.

**Antibodies**. The following primary antibodies were used: Chicken anti-MAP2 (antibodies Online; ICC: 1:1000); guinea pig anti-Shank3 (Synaptic Systems # 162 304; ICC: 1:500); rabbit monoclonal anti Dcp1a, Abcam #183709; ICC 1:1000); rabbit anti-Shank and anti-mRFP antisera have been generated by custom immunization by Biogenes GmbH, Berlin, Germany. Anti-Dicer antibody (Bethyl, 1:1000), anti-HA antibody (Covance Research Products, clone 16B12, 1:1000); anti-AGO2 (EMD Millipore; clone 11A9; #MABE253; 1:1000); and anti-TNRC6ABC, clone 7A9 (Merck Millipore) were used for protein detection in Western Blotting. Secondary antibodies: Alexa fluor 633 goat anti-rb, Alexa 405 goat anti-chk IgG (abcam), Abberior star red goat anti guinea pig were used at 1:1000 dilution.

Cell lines and transfections. HEK293T, U2OS and HeLa cells were obtained from ATCC and cultured in Dulbecco's modified Eagle medium (DMEM; ThermoFisher) supplemented with 10% fetal bovine serum (FBS; GE Healthcare) and penicillin-streptomycin (100 U/mL and 100 mg/mL, respectively; ThermoFisher) under 5% $CO_2$ and at 37 °C. For deletion of the endogenous *AGO2* gene, cells were transfected with a Crispr targeting construct in pLentiCrisprV2 (Genscript), carrying a guide sequence encompassing the *AGO2* start codon (seq.: 5′-GCCACCATGTACTCGGGAG-3′). Cells were selected with puromycin (2 μg/ml) for several days. Absence of AGO2 expression was verified by Western blotting. For silencing assays, cells were plated on 12 well dishes and transfected 4–6 h later using Turbofect transfection reagent (3.5 μl/well). Cells were transfected with an expression vector for a gene of interest (mRFP-Shank3, mRFP-DDX1, or GFP-tagged δ-catenin) in combination with an shRNA vector targeting the corresponding mRNA. For this, shRNA constructs were generated in pSuper for DDX1 and δ-catenin and in pLVTHM for Shank3 using target sequences AGGAGGAGGACCTGATAAA (rat DDX1; bp 1633–1651 in NM_053414.1), GCAACTATGTCGACTTCTA (mouse δ-catenin; 4240–4258 in NM_008729.3) and GGAAGTCACCAGAGGACAAGA (rat Shank3; 3794–3814 in NM_021676.2). For each well either empty vector or AGO2-expression vectors were added to the transfection mix (1 μg/well).

Primary human dermal fibroblast cultures were established from skin biopsies taken from the three patients (cases 2,3 and 13) and four age-matched controls (controls 1, 2, 3, and 5; aged 2, 2, 2, and 15 years at sampling, respectively). In addition, control 4 (GM01887 aged 7 years at sampling) was obtained from Coriell Institute. Primary fibroblasts were cultured in the same manner as the cell lines mentioned above.

Primary dissociated hippocampal neurons isolated from embryonic (E18) rats were co-transfected after 7 days in vitro (DIV7) with GFP-AGO2 constructs (in the FUW vector) and pmRFP-C1 using the calcium phosphate method[52]. The neurons were fixed at DIV14 and stained for endogenous Shank3 (postsynaptic marker); MAP2 (dendritic marker) and DCP1a (P-body marker). RFP fluorescence was used to verify successful transfection. All animal experiments were approved by, and conducted in accordance with, the guidelines of the Animal Welfare Committee of the University Medical Center (Hamburg, Germany) under permission number Org766.

**Microscopy**. Confocal images of hippocampal neurons were acquired with a Leica Sp5 confocal microscope using a ×63 objective. Quantitative analysis for images was performed using ImageJ. Three independent experiments were performed for all neuron data. Primary dendrites were counted at a ring within 10 μm distance from the cell body. 12–15 neurons per each condition were counted. For counting postsynaptic Shank3 clusters, 12–15 neurons per each condition with a total of 45 branches were evaluated. For counting dendritic AGO2-clusters, these were counted along entire dendritic branches. Cluster density was obtained by dividing the number of AGO2 clusters/P-bodies by μm of dendrite length. 12–15 neurons with a total of 36–45 dendrites per each condition were evaluated.

**Immunoprecipitation (IP)**. For immunoprecipitation of overexpressed FLAG/HA-tagged AGO2 proteins, anti-FLAG M2 affinity agarose gel (Sigma-Aldrich) was used and washed twice with cold PBS before incubation with lysate. After incubation of 2.5 h at 4 °C on a rotating wheel, the beads with bound proteins were centrifuged for 1 min at 1000 × g and the supernatant was removed. For qRT-PCR analysis, the affinity matrix was washed with NET buffer (50 mM Tris/HCl pH 7.5, 5 mM EDTA, 0.5% NP-40, 10% Glycerol, 1 mM NaF, 0.5 mM DTT, 1 mM AEBSF) + 300 mM NaCl twice, once with lysis buffer + 450 mM NaCl, once with 600 mM NaCl, once with 450 mM NaCl, followed by washing with PBS once. For Western Blot, Northern Blot and MS analysis, beads were washed three times with NET-lysis buffer with 300 mM NaCl and once with PBS. Subsequent mass spectrometric analysis was performed as described below[30].

Targeted quantification of AGO2 phosphorylation by SRM (Selected Reaction Monitoring). Phosphorylation levels of AGO2 variants were quantified by obtaining selected AGO2 phosphopeptides as well as their non-phosphorylated counterparts as stable isotope-labeled and quantified spike-in standards from JPT (Innovative Peptide solutions, Berlin) f[30]. The following $^{13}C^{15}N$-labeled peptides: SASFNTDPYVR and SApSFNTDPYVR for detection of phospho-S387; YHLVDKEHDSAEGSHTSGQSNGR and YHLVDKEHDpSAEGSHTSGQSNGR for detection of pS824; and YHLVDKEHDpSAEGpSHTpSGQpSNGR for detection of the pS824/pS828/pS831/pS834 cluster, were used to set up a SRM method on a hybrid triple quadrupole/linear ion trap instrument (QTRAP4500, SCIEX). A spectral library built from DDA (data-dependent analysis) runs of the heavy

peptides was built and imported into the open source software Skyline (MacCoss lab software, Seattle, USA). In Skyline, then a targeted method was built according to the occurrence of precursor charge states +2, +3, +4 during several DDA runs. After manual inspection of MS2 spectra at least 3 transitions were selected for each peptide. The resulting transition list was imported into the instrument software (Analyst 1.6.1) and the following parameters were set: Q1 and Q3 at unit resolution (0.7 m/z half-maximum peak width), dwell time 20 ms, cycle time < 3 s. After annotating peptide retention times from the initial SRM run and setting the following parameters: cycle time: 2 s, retention time window: 5 min, a scheduled SRM method was created in Skyline. Sample preparation of overexpressed and immunoprecipitated Flag/HA-tagged Ago2 variants was performed as follows: after separation on SDS-PAGE AGO2 bands were excised, washed with 50 mM $NH_4HCO_3$, 50 mM NH4HCO3/acetonitrile (3/1), 50 mM NH4HCO3/acetonitrile (1/1) and lyophilized. AGO2 variant proteins were then subjected to overnight in gel tryptic digest at 37 °C with ~2 μg trypsin per 100 μl gel volume (Trypsin Gold, mass spectrometry grade, Promega). Importantly, 100 fmol of each heavy peptide were spiked into the digests. After digestion, peptides were extracted twice with 100 mM NH4HCO3, followed by 50 mM NH4HCO3 in 50% acetonitrile. The combined eluates were lyophilized and reconstituted in 20 μl 1% TFA for LC-MS analysis. The LC-MS/MS system consisted of an UltiMate 3000 RSLCnano System (Thermo Scientific, Dreieich) coupled via a NanoSpray II source (SCIEX) to a QTRAP4500 mass spectrometer. Peptides were separated by reversed-phase chromatography on an Acclaim Pepmap100 C18 nano column (75 μm i.d. × 150 mm, Thermo Fisher) with a C18 Acclaim Pepmap100 preconcentration column (100 μm i.d. × 20 mm, Thermo Fisher) in front. At a flow rate of 300 nl/min, a 60 min linear gradient of 4–40% acetonitrile in 0.1% formic acid was used. SRM measurements resulted in.wiff files which were imported back into Skyline. By calculating the heavy-to-light ratios of the peak areas of the respective transitions absolute quantification of endogenous phosphorylated or non-phosphorylated peptides was facilitated. Relative quantification of phosphorylated AGO2 peptides was performed in Excel by first calculating the absolute amount of either peptide species, followed by adding up the amounts of the non-modified peptide species and the related phosphorylated peptide species. Assuming this sum to represent 100%, it was possible to calculate the percentage of the individual phosphopeptide species.

**Quantitative real-time PCR (qRT-PCR)**. For qRT-PCR analysis of overexpressed FLAG/HA-tagged AGO2 proteins, the affinity matrix after immunoprecipitation was washed with lysis buffer + 300 mM NaCl in total twice, once with lysis buffer + 450 mM NaCl, once with 600 mM NaCl, once with 450 mM NaCl, followed by washing with PBS once. The RNA of Input and IP samples were isolated using TRIzol (Thermo Fisher Scientific) and a second step with chloroform. For cDNA synthesis, 1 μg of the Input and complete RNA yield of the IP samples were first digested with DNaseI (Thermo Fisher Scientific). After the digest, cDNA was synthesized using the First-Strand cDNA synthesis kit (Thermo Fisher Scientific), following the manufacturer's protocol. qRT-PCR was performed with Sso Fast Eva Green Mix (Bio-Rad). *NFIC*, fwd 5′-GACCTGTACCTGGCCTACTTTG, rev 5′-CACACCTGACGTGACAAAGCTC; *F8A1*, fwd 5′-GTTTGCGTCTGGGGA GGAAT, rev 5′-TGGTAACGTTCAGCCAACGA; *HAND1*, fwd 5′-GGAGTCC GCAGAAGGGTTAAA, rev 5′-CGGGCAAGGCTGAAAATGAG; *HOXC8*, fwd 5′-CGGAGACGCCTCCAAATTCT, rev 5′-GCCTTGTCCTTCGCTACTGT. qRT-PCRs were run on a CFX96 cycler (Bio-Rad) and data were analyzed using ΔΔCt method[30].

For family survey of case 20, qRT-PCR analysis of *AGO2* and *PTK2* mRNA was performed using the kit LightCycler® 480 SYBR Green I Master following the manufacturer's protocol (Roche), with detection on a Roche LightCycler 480 Real-Time PCR instrument (Roche Diagnostics Corporation, USA). *SULF1* was used as reference gene for normalization. The primer pairs were: *AGO2*, fwd 5′-GATATG CCTTCAAGCCTCCA, rev 5′-AACTCTCCTCGGGCACTTCT; *PTK2*, fwd 5′- TG GGTGAGCTCATCAACAAG, rev 5′- GCCCAAGCATTTTCAGTCTT; *SULF1*, fwd 5′- CCCCCAAGAAATGGTCACTA, rev 5′- CAGGCAAGACTGCCCTA GAC. Data were analyzed using ΔΔCt method.

**In vitro cleavage assay**. The cap-$^{32}$P-labeling of target RNA perfect complementary to the endogenous miR-19b, and the in vitro cleavage assay were performed according to described protocols[53,54]. For this, 25% of the total immunoprecipitate was separated for subsequent analysis by Western Blotting. Thereafter, translation mix was added to a final concentration of 1 × translation mix and the reaction was started by addition of cap-labeled target RNA. The reaction was incubated for 1 h at 30 °C and stopped by addition of TRIzol (Thermo Fisher Scientific) and chloroform, shaking and centrifuging, followed by precipitation overnight at −20 °C in ethanol with 20 μg glycogen RNA grade (Thermo Fisher Scientific). After pelleting, the RNA was resuspended in RNA sample buffer.

**RNA sequencing and gene expression analysis of primary fibroblasts**. Total RNA was extracted with the RNAeasy mini kit (Qiagen) from primary fibroblasts that were in all in the same passage 8. These included primary fibroblasts of three patients bearing *AGO2* mutation (cases 2, 3, and 14), 4 age-matched individuals affected by NDD not related to *AGO2* mutation, and an apparently healthy

individual control 4 (GM01887 aged 7 years at sampling) obtained from Coriell Institute. RNA integrity and quality was assessed with Epoch Microplate Spectrophotometer (BioTek) and on 1% agarose gels. RNA purity was checked using the NanoPhotometer® spectrophotometer (IMPLEN, CA, USA). RNA concentration was measured using Qubit® RNA Assay Kit in Qubit® 2.0 Flurometer (Life Technologies, CA, USA). RNA integrity was assessed using the RNA Nano 6000 Assay Kit of the Bioanalyzer 2100 system (Agilent Technologies, CA, USA). Library preparation and transcriptome sequencing were performed at Novogene. In brief, a total amount of 3 μg RNA per sample was used as input material for the RNA sample preparations. Sequencing libraries were generated using NEBNext® Ultra™ RNA Library Prep Kit for Illumina® (NEB, USA) following the manufacturer's recommendations and index codes were added to attribute sequences to each sample. mRNA was purified from total RNA using poly-T oligo-attached magnetic beads. Fragmentation was carried out using divalent cations under elevated temperature in NEBNext First-Strand Synthesis Reaction Buffer (5× first-strand cDNA was synthesized using random hexamer primer and M-MuLV Reverse Transcriptase (RNase H-. Second strand cDNA synthesis was subsequently performed using DNA Polymerase I and RNase H. Remaining overhangs were converted into blunt ends via exonuclease/polymerase activities. After adenylation of 3′ ends of DNA fragments, NEBNext Adaptor with hairpin loop structure was ligated to prepare for hybridization. In order to select cDNA fragments of preferentially 150~200 bp in length, the library fragments were purified with AMPure XP system (Beckman Coulter, Beverly, USA). Then 3 μl USER Enzyme (NEB, USA) was used with size-selected, adaptor-ligated cDNA at 37 °C for 15 min followed by 5 min at 95 °C before PCR. Then PCR was performed with Phusion High-Fidelity DNA polymerase, Universal PCR primers and Index (X) Primer. At last, PCR products were purified (AMPure XP system) and library quality was assessed on the Agilent Bioanalyzer 2100 system. The clustering of the index-coded samples was performed on a cBot Cluster Generation System using HiSeq PE Cluster Kit cBot-HS (Illumina) according to the manufacturer's instructions. After cluster generation, the library preparations were sequenced on an Illumina Hiseq platform and 125 bp/150 bp paired-end reads were generated. Quality trimming and adapter cutting were performed using Cutadapt v2.5. Genome mapping in paired-end mode was done using Bowtie2 v2.3.4.1 to the human genome GRCh38.p12. Read counts were obtained using bedtools v2.26.0 summarized per protein-coding gene using annotation version GRCh38.97.

Differential expression (DE) analysis was performed by calculating expression fold-changes for each gene in patients and compared to the age-matched controls. To account for high variations among (human) individuals we set a threshold of twofold difference to extract DE genes. Gene ontology enrichment analysis for molecular function and biological process were obtained using the DAVID tool v6.8.

**Molecular dynamics simulations**. WT-AGO2 and AGO2 variants were simulated in the apo-AGO2 state and in four AGO2-RNA complexes using classical all-atom MD (Supplementary Fig. 13). In total, 60 non-biased MD trajectories were obtained; the minimal trajectory length in this set is 200 ns, with a cumulative simulation time of ~17 μs. All MD simulations were conducted in NAMD 2.12[55] using CHARMM36 force field[56] in TIP3P[57] water box. Production MD simulations were performed in NVT ensemble using Langevin thermostat at standard conditions with 2 fs integration step. Principal component analysis of Cartesian coordinates of protein Cα atoms was applied to obtain dominant conformational modes using ProDy[58]. To induce helix7-mediated duplex unwinding and to probe helix7 movements upon p.L192P mutation and perturbed guide-PAZ interactions, we performed metadynamics (MetD) simulations (reviewed in ref. [59]). In the case of non-tempered MetD of WT-AGO2 and p.L192P in the complex with a mismatched duplex, we aimed at inducing g6-g7 kink by enhancing sampling of I365 intercalation between g6 and g7. The sampling of the collective variable (I365δ-g (6,7)) was not restrained, leading to sampling outside the grid for Gaussian potentials deposition. Thus, the potential of mean force (PMF) from this set of MetD simulations cannot be used. Trajectories from not fully converged MetD simulations can be used to obtain mechanistic insights but with some restrictions as discussed in the original reference[60]. The setup and performance of all simulations are described in detail in Supplementary Methods. Colvar module of NAMD was used for enhanced sampling simulations[61].

**Statistical analyses**. Statistical analyses were performed with Prism software (GraphPad, San Diego, CA).

**Reporting summary**. Further information on research design is available in the Nature Research Reporting Summary linked to this article.

## Data availability

The RNA-seq data were submitted to GEO repository under the accession number GSE141099. The mass spectroscopy data for quantification of AGO2 phoshorylation have been submitted to the peptideatlas repository under accession PASS01561. The raw whole-exome sequencing and microarray-based comparative genomic hybridization data that support the findings in affected individual cannot be made publicly available for reasons of patient confidentiality. Qualified researchers may apply for access to these data, pending institutional review board approval. Cells are available upon signing a material transfer agreement. The data supporting the findings of this study are available from the corresponding authors upon reasonable request. Source data are provided with this paper.

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

## Acknowledgements

We thank all family members for their participation and collaboration, Hans-Hinrich Hönck (Institute for Human Genetics, UKE Hamburg) for technical assistance, and UKE microscopic imaging facility (umif) for providing assistance with confocal microscopes. This work was funded in part by Werner Otto Stiftung (to D.L and H-J.K), Deutsche Forschungsgemeinschaft (LE4223/1-1 to D.L.; Kr1321/8-2 to H-J.K; KI 488/7-2 to S.K.; SFB960/B3; DFG/ANR 2064/7-1 to G.M.), postdoctoral fellowships from For-schungsförderungsfonds Medizin at UKE Hamburg (to F.H.N. and V.M.), the DAAD fellowship (to A.K.), the Clinician Scientist Program Medizinische Fakultät der Universität Leipzig (to D.L.D.), the Dietmar-Hopp-Stiftung (to S.S, 23011236), the National Institute of Neurological Disorders and Stroke (NINDS) under award number K08NS092898, Jordan's Guardian Angels, the Brotman Baty Institute for Precision Medicine (to G.M.M.), the SOLVE-RD program (EU to H.G.B.), the National Research Foundation Singapore under its National Medical Research Council Centre Grant Pro-gramme (Project No. NMRC/CG/M003/2017 to E.C.T.), and, in part, under the Car-e4Rare Canada Consortium funded by Genome Canada and the Ontario Genomics Institute (OGI-147), the Canadian Institutes of Health Research, Ontario Research Fund, Genome Alberta, Genome British Columbia, Genome Quebec, and Children's Hospital of Eastern Ontario Foundation (to K.M.B.). T.R., K.R.D. and C.A.E. are supported through the Australian NHMRC Centre for Research Excellence in Neurocognitive Disorders.

## Author contributions

D.L., D.M.Z, F.H.N., A. Bruckmann, V.G., I.M., V.M., I.L., C.S., S.K., Z.I., G.M., and H.-J.K. performed cell experiments and analyzed the data. D.L., M.R.F.R., M.M., B.L., E-S.T., E.G., J.J., J.D., E.Z-H., T. Kovacevic, L.R., K.F., D.M., S.S., A.M., M.S., B.P., J.L., T.B., C.M., P.P., S. Lüttgen, J.P., R.R., M.W., T.G., K.L., P.R., H.G.B., C.C., S.L., D.A.D., K.M.B., G.M.M., A.M-R., T.R., P.I.A., and K.R.D were involved in patient care and gathered detailed clinical information for the study. D.L. analyzed clinical data. D.L., M.R.F.R., E-S.T, A.T., H.H.L., B.W.C., D.L.D., M.O., T.H., J.M., R.A.J., A.P.A.S., C.K., E.-C.T., G.M.M., K.M., T.Kl., R.P., C.A.E., K.R.D., and T.R. analyzed and supported the trio-exome sequencing results. A.K. and Z.I. performed non-biased molecular dynamics (MD) simulations. D.L., A. Bartholomäus and Z.I. analyzed the RNA-sequencing data. D.L., Z.I., G.M., and H-J.K. directed functional analyses. D.L. and H-J.K. jointly directed the study and drafted the initial manuscript.

## Funding

## Competing interests

A.T. and K.M. are employees of GeneDx, Inc. The other authors declare that they have no conflict of interest.

## Additional information

Davor Lessel [1✉], Daniela M. Zeitler[2], Margot R. F. Reijnders [3,4], Andriy Kazantsev[5], Fatemeh Hassani Nia [1], Alexander Bartholomäus [5,6], Victoria Martens[1], Astrid Bruckmann[2], Veronika Graus[2], Allyn McConkie-Rosell[7], Marie McDonald[7], Bernarda Lozic[8,9], Ee-Shien Tan[10], Erica Gerkes[11], Jessika Johannsen[12], Jonas Denecke[12], Aida Telegrafi[13], Evelien Zonneveld-Huijssoon [11], Henny H. Lemmink [11], Breana W. M. Cham[10], Tanja Kovacevic [8], Linda Ramsdell[14], Kimberly Foss[14], Diana Le Duc [15], Diana Mitter[15], Steffen Syrbe[16], Andreas Merkenschlager [17], Margje Sinnema[4], Bianca Panis[18], Joanna Lazier[19], Matthew Osmond[20], Taila Hartley[20], Jeremie Mortreux [21,22], Tiffany Busa[21], Chantal Missirian[21,22], Pankaj Prasun[23], Sabine Lüttgen[1], Ilaria Mannucci [1], Ivana Lessel[1], Claudia Schob[1], Stefan Kindler[1], John Pappas [24], Rachel Rabin[24], Marjolein Willemsen[3], Thatjana Gardeitchik[3], Katharina Löhner[11], Patrick Rump [11], Kerith-Rae Dias [25,26], Carey-Anne Evans[25,26], Peter Ian Andrews[27,28], Tony Roscioli[25,29,30], Han G. Brunner[3,4], Chieko Chijiwa[31], M. E. Suzanne Lewis[31], Rami Abou Jamra [15], David A. Dyment[19,20], Kym M. Boycott[19,20], Alexander P. A. Stegmann [3,4], Christian Kubisch[1], Ene-Choo Tan [32], Ghayda M. Mirzaa [33,34,35], Kirsty McWalter[13], Tjitske Kleefstra[3], Rolph Pfundt[3,11], Zoya Ignatova[5], Gunter Meister[2] & Hans-Jürgen Kreienkamp [1✉]

[1]Institute of Human Genetics, University Medical Center Hamburg-Eppendorf, 20246 Hamburg, Germany. [2]Regensburg Center for Biochemistry (RCB), Laboratory for RNA Biology, University of Regensburg, Regensburg, Germany. [3]Department of Human Genetics, Radboud University Medical Center, 6500 HB Nijmegen, The Netherlands. [4]Department of Clinical Genetics, Maastricht University Medical Center, Maastricht, The Netherlands. [5]Institute of Biochemistry & Molecular Biology, University of Hamburg, Hamburg, Germany. [6]GFZ German Research Centre for Geosciences, Section Geomicrobiology, Potsdam, Germany. [7]Division of Medical Genetics, Department of Pediatrics, Duke University, Durham, NC 27707, USA. [8]University Hospital of Split, Split, Croatia. [9]University of Split School of Medicine, Split, Croatia. [10]Genetics Service, Department of Paediatrics, KK Women's & Children's Hospital, Singapore, Singapore. [11]Department of Genetics, University of Groningen, University Medical Center Groningen, Groningen, The Netherlands. [12]Department of Pediatrics, University Medical Center Eppendorf, 20246 Hamburg, Germany. [13]GeneDx, Gaithersburg, MD 20877, USA. [14]Division of Genetic Medicine, Seattle Children's Hospital, Seattle, WA 98105, USA. [15]Institute of Human Genetics, University of Leipzig Hospitals and Clinics, Leipzig, Germany. [16]Department of General Paediatrics, Division of Pediatric Epileptology, Centre for Paediatrics and Adolescent Medicine, University Hospital Heidelberg, Heidelberg, Germany. [17]Department of Neuropediatrics, University of Leipzig, Leipzig, Germany. [18]Department of Pediatrics, Zuyderland Medical Center, Heerlen and Sittard 6419, the Netherlands. [19]Department of Genetics, Children's Hospital of Eastern Ontario, Ottawa, ON, Canada. [20]Children's Hospital of Eastern Ontario Research Institute, University of Ottawa, Ottawa, ON, Canada. [21]Département de Génétique Médicale, CHU Timone Enfants, Assistance Publique - Hôpitaux de Marseille AP-HM, Marseille, France. [22]Aix Marseille Univ, INSERM, MMG, U1251 Marseille, France. [23]Department of Genetics and Genomic Sciences, Icahn School of Medicine at Mount Sinai, New York, USA. [24]Department of Pediatrics, New York University Grossman School of Medicine, New York, NY 10016, USA. [25]Neuroscience Research Australia (NeuRA), Prince of Wales Clinical School, University of New South Wales, Sydney, Australia. [26]NSW Health Pathology Randwick Genetics, Sydney, Australia. [27]Department of Neurology, Sydney Children's Hospital, Sydney, Australia. [28]School of Women's and Children's Health, University of New South Wales, Sydney, Australia. [29]Centre for Clinical Genetics, Sydney Children's Hospital, Sydney, Australia. [30]New South Wales Health Pathology Genomics Laboratory Randwick, Sydney, Australia. [31]Department of Medical Genetics, University of British Columbia, Vancouver, BC V6H 3N1, Canada. [32]Research Laboratory, KK Women's & Children's Hospital, Singapore, Singapore. [33]Center for Integrative Brain Research, Seattle Children's Research Institute, Seattle, WA, USA. [34]Department of Pediatrics, University of Washington, Seattle, WA, USA. [35]Brotman Baty Institute for Precision Medicine, Seattle, WA 98195, US. ✉email: d.lessel@uke.de; kreienkamp@uke.de

