## [Peer Review File · Nature Communications]

Reviewers' comments:

Reviewer #1 (Remarks to the Author):

Lessel and colleagues identify germline AGO2 mutations associated with impaired neurological development in humans. They perform MD simulations and a variety of cellular and biochemical assays to assess the functional consequences of the identified mutations. From the data the authors suggest that most mutations impair the ability of Ago2 to release bound target RNAs, providing a plausible molecular mechanism for the observed neurological phenotypes. As a basic researcher in the small RNA field, I find study paper exciting and fascinating.

My critique of the study is limited to my expertise—the structural and biochemical analyses—and does not address the validity of the genetic analyses. From a structural and mechanistic perspective, all of the identified mutations make sense and the proposed mechanism is reasonable. I have the following questions/concerns/suggestions:

- 1) Why do the relative levels of Ago2 mutants vary so much between blots in Fig. 2A? Did the authors perform any loading controls? The variability in Ago2 levels between samples on the blot is worrisome because the silencing activity being assessed likely correlates with the amount of Ago2 in the cell.
- 2) Why are there no p-values associated with the data in Fig.3A-B?
- 3) In the MD simulations, it immediately makes sense that the C1'–C1' distance at positions 5-7 of the guide-target duplex would increase as the α H7-MID distance decreases, but I would not have expected the same for positions 2 and 3. Does the C1'–C1' distance change uniformly at all positions in the guide-target duplex, or are nucleotides closer to α H7 more sensitive to its position?
- 4) Can the authors include more detail about how the 20 patients were identified in the main text? As it currently reads, the manuscript suggests to me that more than half of all people affected by “mild to severe global neurodevelopmental delay” carry AGO2 mutations. Can this possibly be true? Or, were additional indicators/selections used identify members of the cohort?
- 5) Consider noting that mutations near the L1 hinge (such as p.F182del, p.G201C, p.G201V, and p.H203Q) have been shown to inhibit unwinding of siRNA duplexes (see PMID: 22233755).
- 6) Consider noting that mutations p.G733R, p.C751Y, and p.S760R almost certainly disrupt the local structure of the Ago2 seed-binding region and thereby likely compromise display of the seed for target-pairing. Also, the p.G733R mutation is especially severe in terms of steric clashes and likely causes substantial misfolding.
- 7) The term “RISC complex” is often used in the text but is redundant because the C is RISC stands for “complex”. Consider changing to RISC.

8) Line 164: “reduced phosphorylation of the C-terminal serine cluster of most of the disease-causing mutants results in reduced target release and thus an extended dwelling time of the AGO2 mutants on their targets.” Considering that 1) none of the mutations are found in C-terminal cluster, and 2) several mutations are predicted to disrupt helix-7 in a manner known to reduce target release rates, the reverse possibility should be considered: i.e. mutations result in reduced target release rates, leading to reduced phosphorylation levels in the C-terminal serine cluster.

I congratulate the authors on a fascinating and important study.

Best wishes,
Ian J. MacRae

Reviewer #2 (Remarks to the Author):

I enjoyed reviewing this paper, and many of us have been waiting for a human AGO2 phenotype to be reported. I would like to see this paper published, and the authors have collected a significant cohort of affected individuals for what is assumed to be a very rare condition, and performed excellent functional biology studies to prove and explain the pathogenicity of the mutations they have found. My comments are aimed to help improve and clarify the paper, and should not be taken as criticisms. My main suggestion is the improvement of the phenotype description – so the paper can be of use to physicians.

Abstract

Last sentence needs to be clearer and could be improved.

Main text

Para1:

Refs 2 and 3 are from 2005 papers and do not justify or refer to data on “each miRNA may recognise a set of target mRNAs... Could the sentence be better referenced please?

...outmost importance for the proper development.... (? utmost)

“Novel monogenic disorder” is it a phenotype or not, if not then it isn’t a disorder as this infers a clinical identity. So, can the authors justify this as a coherent phenotype, or not.

Para2:

Did the authors really just sequence 20 patients – what was the real method of ascertainment?

The second sentence is incomplete.

Para3:

This paper will be used by clinicians caring for children with all types of developmental delays – and the data within this paper should be of use to them. The presentation of the clinical data is woefully bad, and this needs substantial improvement. A Table of case, age, sex, gene mutation, protein mutation,

proposed mutation mechanism, and some summary phenotype would be very useful in the main text. “issues” is an inappropriate word and should be changed.

A summary of the phenotypic features by incidence is needed in the main text and not just a partial list which appears to be in a ransom order.

“We name this condition Argonaute-2 (AGO2) syndrome” – this is the decision of a HUGO nomenclature committee, whom the authors should contact. Also, this seems a very broad phenotype and if there are distinctive or obligatory features the authors should state this. I would have like far greater attention to phenotype and phenotype-genotype than the authors have given.

Looking at the extended data I bring only two examples to show the apparent lack of attention sown by the authors to the human phenotypes of those with AGO2 mutations; there are more and this whole section needs attention. 1) This case needs more medical data, and the sentence “and head circumference of 46.5 cm (-0.7 SD). He had relative macrocephaly, ...” is unfortunately laughably poor. 2) Case three has mosaic Pallister Killian syndrome – which is not further mentioned nor discussed as to what elements contribute to the phenotype. Also a Table of phenotypic features needs adding. Finally, the deletion case that involves both AGO2 and PTK2 – no mention is made of the contribution of PTK2 to the phenotype – it too is a gene with a pLI of 1!

Para4:

“AGO2 consists of N, PAZ, MID and PIWI domains which are connected by linker regions L1 and L2.” I am not sure this sentence contributes in this position. Far better would have been an introductory sentence explaining the paragraphs function, or aims. If the authors do want to use this sentence then please explain each domains name and function. Figure 1 does not give the domain information either, and the colours used (on my monitor) didn’t correlate between the 3D figure and the annotated linear peptide figure. Would the authors like to speculate/hypothesise why p.G573S causes dysfunction? Finally the deletion case – referencing the paper about 5 children with combined AGO1 and AGO3; surely the authors should compare and contrast their cases to these (rather milder, and oddly looking rather like velocardiofacial syndrome).

AGO2 and shRNA silencing para:

I do not feel qualified to assess this work, but reverse order of sentence 2 would aid readability. I think the last sentence needs more justification – were the mutation specific effects due to altered protein stability, or some other mechanisms?

RISC formation para:

Some comment on why the p.G733R mutation is different to the others reported would be welcome.

Molecular dynamics paragraph

I do not feel qualified to comment on this.

Global transcription paragraph:

The authors should comment on the difficulties of using fibroblast data to extrapolate to neural and glial cell functions. I don’t think referencing a kinetochore paper is useful here – the data stands as it is, and strongly suggests that AGO2 mutations would very likely affect many neuronal genes – and it may also

be useful to inform the readers about the phenotype of the AGO2 deficient fibroblasts, i.e. did they grow normally or have obvious mitotic defects?

Concluding paragraph (unmarked as such):

This should be reversed: “In line with the extreme intolerance of AGO2 to mutations in humans in general, our analyses suggest that even subtle changes in AGO2 function are sufficient to elicit aberrant neurological development as observed in AGO2 syndrome. “ Also, some authors may misunderstand the words “extreme intolerance of AGO2 to mutations..”, and it should be clear that this refers to a lack of mutations found in population databases, not cell biology experiments?

Argonaute should be ARGONAUTE if the authors are referring to the human protein?

A concluding sentence is missing.

Figure 1

Enlargement isn't as indicated by the dotted box, and has a different orientation. I had difficulty distinguishing the blues from each other – so could this be clear in the final paper?

Extended data

I couldn't find any methods concerning the AGO2 antibody used in Extended Data Figure 5, nor why it is called alpha AGO2.

In Extended Data Figure 10, eyeballing the SHANK3 expression data versus the bar graph of SHANK3 density does not seem to correlate – could this be clarified, explained or fixed?

Reviewer #3 (Remarks to the Author):

In this manuscript, Lessel et al. demonstrated the mutations in germline AGO2 affects the function of RISC, in particular for the neurological development by identifying 13 mutations in 20 patients with neurological development diseases. The authors used multi-omics level analysis including quantitative mass spectrometry, which this review will focus on.

The quantitative mass spectrometry was used in targeted fashion to measure the phosphorylation level of Ser387 and C-term cluster upon immunoprecipitation of AGO2. The authors argue (with the results shown Fig 3a) that the phosphorylation of C-term cluster was reduced whereas Ser387 phosphorylation is not changed significantly except for H203Q mutation.

However, this analysis is not persuasive as it is unless more information from two major sources are provided.

First, I failed to find the mass spectrometry related method detail to generate the results in Fig 3. The authors are referring to the previous paper (Phosphorylation of Argonaute proteins affects mRNA

binding and is essential for microRNA-guided gene silencing in vivo) for the method detail, but the referring paper seems not to contain the MS instrumental set-ups, parameters, or protocols. Did the authors use normal shotgun proteomics or targeted method such as SRM or MRM? Even if this information was provided (and I failed to find) in the previous paper, it should be again provided in this manuscript even briefly.

Second, how the ratio between p vs np was measured is not clear. Did the authors run automated tools or was it done manually? How the phospho peptides were identified? Such information could be useful for readers to understand the results and will make this manuscript more convincing.

Minor points for Fig 3a:

- 1) The y-axis reads "ratio p/n.p. peptide (%)". In legends, please specify p/n.p means phospho vs. non-phospho. Also please explain why the unit is %. The ratio should have no unit in general.
- 2) The meaning of whisker in bar plot is not shown. In case of H203Q, the whisker is rather wide; it is not clear this ratio bar has statistical meaning. Please calculate p-value for each case and mark the statistically meaningful ones with *, for example. Provide how the p-values are calculated.
- 3) The axis font is too small.
- 4) Please use the term quantitative mass spectrometry instead of mass spectroscopy.

Reviewer #4 (Remarks to the Author):

In their manuscript "Germline AGO2 mutations impair RNA interference and human neurological development", Lessel et al revealed 13 heterozygous mutations in Ago2 gene in 20 patients with neurological disorders. Although these are heterozygous and do not affect known Ago2 functional domains or modification sites, they affect conserved residues, many are somewhat recurrent (in the sense that 5 of them occurred as independent de novo events), and several are in silico predicted as pathogenic.

Based on this, they conduct assays of the variant Ago2 molecules, primarily by expressing these in Ago2^{-/-} cells or in wildtype primary cells. From these tests, they provide evidence that these mutations can affect the function of Ago2 in different levels. For example, they concluded from their assays that individual single amino acid mutations can impair shRNA-mediated silencing, reduce phosphorylation that may be important for target release, or increase formation of dendritic P-bodies.

They also analyzed transcriptome changes in fibroblasts from several Ago2 patients, suggesting some overlaps although it is hard to draw clear direct conclusions to miRNA functional changes from these

tests.

Overall, the clinical significance and dysfunction mechanism for Ago2-mutations revealed by this manuscript make the topic appropriate for Nature Communications. I have the following major concern and a few minor concerns. If they could be addressed, the paper would be a good candidate for publication, but it is somewhat needing further evidence to support the functional conclusion.

Major concern.

It is recognized from mouse studies that ago2 is lethal gene, so we don't expect human homozygous patients. However, I am not aware of reported heterozygous ago2 mouse phenotypes. It is certainly possible the human defects require a sensitive phenotypic setting like the brain/behavior to be revealed. However, the model would have to be for the Ago2 mutant proteins to be directly causal, they need be sufficiently dominant negative to cause a problem in the presence of wildtype Ago2 allele.

It would be best if there were knock-in alleles, but I realize that would require some efforts. Alternatively, the existing data needed to be more robust. Instead, their tests are in the setting of misexpression mutant Ago2 proteins, sometimes in an ago2-knockout setting. With these caveat, they need more rigorous tests to (1) showing altered activities, and (2) if plausible to have effects in the physiological setting.

For example, they concluded all variant Ago2 proteins show decreased shRNA silencing, but they lack several important controls. There can be different amount of transfection efficiency of the constructs, different effective amount of cell lysates loaded, etc. At the very least, they need to redo the experiments with a co-transfected control construct to assess if there is change to the stability of Ago2 proteins or just variable transfections, and to blot another endogenous protein as a loading control for the knockdown efficiency. Also, they should test some kind of miRNA responsive reporter, since the major kind of Ago2 regulation is miRNAs and not target cleavage.

Second, they should do some titration experiment in wildtype cells and compare with blotting of Ago2 (to check total Ago2 proteins) that they are not in a big misexpression range. This can evaluate whether they can still see any functional differences in wildtype vs mutant Ago2. This is useful to know if any effect can be seen in a range that is more like the heterozygous state. For this purpose they can just check a couple of the more promising variants they think are more likely to show a functional readout, doesn't need to be all of them. I am trying to provide a reasonable path they could try to show such mutations can be physiologically relevant, even though they don't have a knockin cell model.

Minor concerns.

1. I am not very familiar with ExAC and gnomAD databases. I thought its very striking that Ago2 is "one

of the most missense-intolerant genes in the human genome" 15th/18000". Indeed, ExAC lists Ago2 at 15th. However, it seems ExAC has been replaced/incorporated into the current gnomAD database, which seems to include more data, and states this is the database to use moving forward. In gnomAD, now Ago2 ranks 818 out of 19703 genes. Can the authors comment on the differences of the databases, and why the older ExAC ranking would be appropriate to cite?

1. In Figure 1B, the authors identified a patient-related deletion of N-terminal of Ago2 due to a 235.5 kb genomic deletion, but this deletion also removed 23 exons of the C-terminal of PTK2 gene which is upstream of Ago2 gene. Does this deletion (if they are in-frame) cause a gene-fusion of PTK2-Ago2 gene, considering many gene fusions can lead to disease? This should be discussed.

2. The y-axis labeling in Figure 4D should be "Ago2 clusters/um dendrite".

3. The Western blotting in Supplementary Figure 8 should be replaced by the full image, but not the cropped one.

4. This is not really a full written paper, it doesn't have a proper introduction and the paper ends very abruptly without discussion or putting into context the relation of these mutations to other proposed disease-associated mutations in Ago2, and other specific Ago2 amino acids that have been suggested from mechanistic studies to impact Ago2 function.

Reviewers' comments:

Reviewer #1 (Remarks to the Author):

Lessel and colleagues identify germline AGO2 mutations associated with impaired neurological development in humans. They perform MD simulations and a variety of cellular and biochemical assays to assess the functional consequences of the identified mutations. From the data the authors suggest that most mutations impair the ability of Ago2 to release bound target RNAs, providing a plausible molecular mechanism for the observed neurological phenotypes. As a basic researcher in the small RNA field, I find study paper exciting and fascinating.

We thank the reviewer for their interest and their thoughtful comments and feedback. Below are our responses to the questions and concerns raised by the reviewer.

My critique of the study is limited to my expertise—the structural and biochemical analyses—and does not address the validity of the genetic analyses. From a structural and mechanistic perspective, all of the identified mutations make sense and the proposed mechanism is reasonable. I have the following questions/concerns/suggestions:

1) Why do the relative levels of Ago2 mutants vary so much between blots in Fig. 2A? Did the authors perform any loading controls? The variability in Ago2 levels between samples on the blot is worrisome because the silencing activity being assessed likely correlates with the amount of Ago2 in the cell.

Response: We have now included new blots, and appropriate loading and transfection controls for each knockdown experiment in Figure 2. These include tubulin, co-transfected mRFP, and GFP expressed from the shRNA silencing vector. Furthermore we have, upon request of reviewer #4, introduced a “titration experiment” where we have used different amounts of Ago2 expression vectors while keeping shRNA constructs and total transfected DNA constant. These experiments (new supplementary Fig. 6) show that the silencing efficiency is, over a wide range of concentrations, independent of the total amount of Ago2. Thus, even at low concentrations (below endogenous levels in 293T cells), wt AGO2 silences a lot more efficiently compared to the mutants (F182del; L192P; G733R).

Please, see also response to comments by reviewer 4.

2) Why are there no p-values associated with the data in Fig.3A-B?

Response: p-values have now been added, showing significant differences for most mutants when compared to wt.

3) In the MD simulations, it immediately makes sense that the C1'–C1' distance at positions 5-7 of the guide-target duplex would increase as the α H7-MID distance decreases, but I would not have expected the same for positions 2 and 3. Does the C1'–C1' distance change uniformly at all positions in the guide-target duplex, or are nucleotides closer to α H7 more sensitive to its position?

Response: MD simulations have now been moved to Fig. 4. In our simulations we see that the C1'–C1' distance does not change uniformly at all positions. The new Fig. 4D summarizes the C1'–C1'

distance measured along the metadynamics trajectories of WT and L192P for positions g2 – g7. The base pairs at positions g2-g3 were not affected by the bias-driven α H7 movements. Reflecting the comment of the Reviewer, we felt that the figures we have included did not explicitly show this non-uniform behavior of the C1'-C1' distance at various positions of the guide-target duplex. Hence, in the revised version, we include a new figure 4D which exemplifies the C1'-C1' distances for the base pairs at the guide positions g2 to g7.

4) Can the authors include more detail about how the 20 patients were identified in the main text? As it currently reads, the manuscript suggests to me that more than half of all people affected by “mild to severe global neurodevelopmental delay” carry AGO2 mutations. Can this possibly be true? Or, were additional indicators/selections used identify members of the cohort?

Response: Please note that we had initially submitted to another Journal of the Nature family, and this information was not explicitly noted in the main text for length constraints. We have now greatly expanded this section to explain how the 20 (actually currently 21) patients were identified:

“During trio whole-exome sequencing of a cohort of 50 children affected by developmental disturbances of unclear etiology and seemingly nonspecific neurological manifestations⁵, we identified a patient bearing a *de novo* missense mutation p.L192P in AGO2 (NM_012154.5). ... These findings motivated us to search for further cases carrying heterozygous AGO2 variants utilizing both the internet-based GeneMatcher tool¹⁰, and direct contact to our network of collaborators.”

5) Consider noting that mutations near the L1 hinge (such as p.F182del, p.G201C, p.G201V, and p.H203Q) have been shown to inhibit unwinding of siRNA duplexes (see PMID: 22233755).

Response: We have consulted the publication mentioned by the Reviewer (Kwak and Tomari; NSMB 19, 2012), in which the authors analyzed 41 mutants within the N and L1 domains of human AGO2 for their ability for siRNA and miRNA duplex unwinding. Interestingly, the authors describe the F181A and F182A mutations, both of which are in the L1 region. We identified one *de novo* mutation that leads to deletion of the F182 (p.F182del). Interestingly, it was shown that both F181A and F182A affect siRNA unwinding, whereas the F181A additionally leads to impaired miRNA unwinding. Reflecting the comment of the Reviewer we have now mentioned these findings in our manuscript. Moreover, we have performed MD simulations for F181A (now in Fig. 4c). As expected, and similar to some of the clinically-related mutations, for F181A we also observed a shift in the α H7-MID distance which is likely related to the AGO2 unwinding function.

6) Consider noting that mutations p.G733R, p.C751Y, and p.S760R almost certainly disrupt the local structure of the Ago2 seed-binding region and thereby likely compromise display of the seed for target-pairing. Also, the p.G733R mutation is especially severe in terms of steric clashes and likely causes substantial misfolding.

Response: As mentioned by the Reviewer, the p.G733R mutation is severe and in our hands exhibited loss-of-function in almost every assay (decreased miRNA binding, target binding, and localization to P-bodies). We have now added a sentence in the Discussion that G733R is likely to cause some misfolding.

By contrast, the p.S760R mutation showed increased target binding (Fig. 3B), thus it is unlikely that this mutation would compromise seed-binding region for pairing, but rather the unwinding ability of AGO2, as also suggested by MD simulations.

7) The term “RISC complex” is often used in the text but is redundant because the C is RISC stands for “complex”. Consider changing to RISC.

Response: We thank this reviewer for careful evaluation of our manuscript. This has been changed accordingly throughout the text.

8) Line 164: “reduced phosphorylation of the C-terminal serine cluster of most of the disease-causing mutants results in reduced target release and thus an extended dwelling time of the AGO2 mutants on their targets.” Considering that 1) none of the mutations are found in C-terminal cluster, and 2) several mutations are predicted to disrupt helix-7 in a manner known to reduce target release rates, the reverse possibility should be considered: i.e. mutations result in reduced target release rates, leading to reduced phosphorylation levels in the C-terminal serine cluster.

Response: We completely agree with the reviewer, and as we don’t know which comes first, target release or phosphorylation, we have changed the words: “...results in reduced target release...” to “...coincides with reduced target release...”

I congratulate the authors on a fascinating and important study.

Response: Thanks a lot, we very much appreciate this comment!

Best wishes,
Ian J. MacRae

Reviewer #2 (Remarks to the Author):

I enjoyed reviewing this paper, and many of us have been waiting for a human AGO2 phenotype to be reported. I would like to see this paper published, and the authors have collected a significant cohort of affected individuals for what is assumed to be a very rare condition, and performed excellent functional biology studies to prove and explain the pathogenicity of the mutations they have found. My comments are aimed to help improve and clarify the paper, and should not be taken as criticisms. My main suggestion is the improvement of the phenotype description – so the paper can be of use to physicians.

Response: We thank the reviewer for their interest and their thoughtful comments and feedback. Please note that we had initially submitted to another Journal of the Nature family. Thus, due to length constraints the phenotype of the cases was only briefly mentioned in the main text. We have now greatly expanded the clinical information in the main text and followed most of the very helpful comments of this reviewer. Below are our responses to the questions and concerns raised by the reviewer.

Abstract

Last sentence needs to be clearer and could be improved.

Response: Following this suggestion we have changed the last sentence to read:

“Our data emphasize the importance of gene expression regulation through the dynamic AGO2-RNA association for human neuronal development.”

Main text

Para1:

Refs 2 and 3 are from 2005 papers and do not justify or refer to data on “each miRNA may recognise a set of target mRNAs... Could the sentence be better referenced please?

Response: We have now added additional references to this paragraph, and to this particular sentence

...outmost importance for the proper development.... (? utmost)

Response: Thank you for the careful reading: Outmost was change to utmost.

“Novel monogenic disorder” is it a phenotype or not, if not then it isn’t a disorder as this infers a clinical identity. So, can the authors justify this as a coherent phenotype, or not.

Response: Indeed, as more comprehensively outlined below, variants in *AGO2* cause a novel genetic disorder characterized by intellectual disability, delayed motor development, impaired speech and receptive language development.

Para2:

Did the authors really just sequence 20 patients – what was the real method of ascertainment? The second sentence is incomplete.

Response: We have now greatly expanded this section to explain how the 20 (actually currently 21) patients were identified:

“During trio whole-exome sequencing of a cohort of 50 children affected by developmental disturbances of unclear etiology and seemingly nonspecific neurological manifestations⁵, we identified a patient bearing a *de novo* missense mutation p.L192P in *AGO2* (NM_012154.5). ... These findings motivated us to search for further cases carrying heterozygous *AGO2* variants utilizing both the internet-based GeneMatcher tool¹⁰, and direct contact to our network of collaborators.”

Para3:

This paper will be used by clinicians caring for children with all types of developmental delays – and the data within this paper should be of use to them. The presentation of the clinical data is woefully bad, and this needs substantial improvement. A Table of case, age, sex, gene mutation, protein mutation, proposed mutation mechanism, and some summary phenotype would be very useful in the main text.

“issues” is an inappropriate word and should be changed.

A summary of the phenotypic features by incidence is needed in the main text and not just a partial list which appears to be in a ransom order.

Response: Following the reviewers suggestion we have now greatly expanded the clinical section and have included the novel table 1, summarizing the major clinical findings. In addition, the paragraph

dealing with description of the patients in the results section has been greatly expanded, and an additional one in the discussion has been added.

“We name this condition Argonaute-2 (AGO2) syndrome” – this is the decision of a HUGO nomenclature committee, whom the authors should contact. Also, this seems a very broad phenotype and if there are distinctive or obligatory features the authors should state this. I would have like far greater attention to phenotype and phenotype-genotype than the authors have given.

Response: Following this reviewer’s suggestion, and despite the fact that this was the explicit wish of some of the patient’s families, we have now refrained from naming this condition.

Looking at the extended data I bring only two examples to show the apparent lack of attention shown by the authors to the human phenotypes of those with AGO2 mutations; there are more and this whole section needs attention. 1) This case needs more medical data, and the sentence “and head circumference of 46.5 cm (-0.7 SD). He had relative macrocephaly, ...” is unfortunately laughably poor. 2) Case three has mosaic Pallister Killian syndrome – which is not further mentioned nor discussed as to what elements contribute to the phenotype. Also a Table of phenotypic features needs adding.

Response: We have now carefully revised the clinical reports in the supplementary data, prepared a new table summarizing the clinical data and additionally discussed the phenotype-genotype relations.

Finally, the deletion case that involves both AGO2 and PTK2 – no mention is made of the contribution of PTK2 to the phenotype – it too is a gene with a pLI of 1!

Response: Indeed, at this point we cannot completely exclude the possibility that the deletion of *PTK2*, in addition to *AGO2*, contributes to the observed phenotype. This, was now clearly added in the revised version.

Para4:

“AGO2 consists of N, PAZ, MID and PIWI domains which are connected by linker regions L1 and L2.” I am not sure this sentence contributes in this position. Far better would have been an introductory sentence explaining the paragraphs function, or aims. If the authors do want to use this sentence then please explain each domains name and function. Figure 1 does not give the domain information either, and the colours used (on my monitor) didn’t correlate between the 3D figure and the annotated linear peptide figure.

Response: We have now made several changes to this paragraph; in particular we have explained domain names and functions. We have also prepared a novel Figure 1a and changed the colours.

Would the authors like to speculate/hypothesise why p.G573S causes dysfunction?

Response: We have added a speculation here that the larger serine instead of small glycine may lead to some local unfolding.

Finally the deletion case – referencing the paper about 5 children with combined AGO1 and AGO3; surely the authors should compare and contrast their cases to these (rather milder, and oddly looking rather like velocardiofacial syndrome).

Response: Indeed, we have noted in the revised version that the case 21 (previous case 20) bearing the deletion which involves the first 3 AGO2 exons, in addition to the last 23 PTK2 exons, somewhat resembles these 5 children bearing combined loss of AGO1&3.

AGO2 and shRNA silencing para:

I do not feel qualified to assess this work, but reverse order of sentence 2 would aid readability. I think the last sentence needs more justification – were the mutation specific effects due to altered protein stability, or some other mechanisms?

Response: We have rewritten the beginning of this paragraph, and hope it is now better readable. Also it has been extended, to accommodate requests by the other reviewers. Finally, we have changed the wording, to be more precise. Now it reads:

“... target mRNA specific effects of some mutations.”

RISC formation para:

Some comment on why the p.G733R mutation is different to the others reported would be welcome.

Response: We have now included a statement saying that this mutation introduces a rather bulky residue, which may lead to some unfolding locally.

Molecular dynamics paragraph

I do not feel qualified to comment on this.

Global transcription paragraph:

The authors should comment on the difficulties of using fibroblast data to extrapolate to neural and glial cell functions. I don't think referencing a kinetochore paper is useful here – the data stands as it is, and strongly suggests that AGO2 mutations would very likely affect many neuronal genes – and it may also be useful to inform the readers about the phenotype of the AGO2 deficient fibroblasts, i.e. did they grow normally or have obvious mitotic defects?

Response: We have included a section in the discussion part where we noted the problem of the differential expression patterns in different organs and tissues. We have also added further references to properly cite the connection between various mitotic defects and neurodevelopmental disorders. Proper and deep analyses of these complex biological processes (cell division, mitotic nuclear division, sister chromatid cohesion, chromosome segregation, microtubule binding and regulation of cell cycle) require extensive further work, and are in our view out of scope for this initial manuscript. We have however started performing some experiments, and our initial data suggest AGO2-fibroblasts do not have severe proliferation defects. However, due to COVID-19 outbreak and partial lab lockdown, we will not be able to complete these experiments in the near future.

Nevertheless, and as explicitly stated in the revised manuscript, these experiments will be the main focus of our future work.

Please also note that during review of our RNA sequencing results we recognized a technical inconsistency in processing of the gene annotation files. Technically a quotation mark character was

ignored leading to block-wise interpretation of several rows as one field. Unfortunately, this led to a large number of missing genes in our first version which is now corrected in the text, tables and figures.

Concluding paragraph (unmarked as such):

This should be reversed: "In line with the extreme intolerance of AGO2 to mutations in humans in general, our analyses suggest that even subtle changes in AGO2 function are sufficient to elicit aberrant neurological development as observed in AGO2 syndrome. " Also, some authors may misunderstand the words "extreme intolerance of AGO2 to mutations..", and it should be clear that this refers to a lack of mutations found in population databases, not cell biology experiments?

Argonaute should be ARGONAUTE if the authors are referring to the human protein?

A concluding sentence is missing.

Response: The concluding paragraph has now been extended into a full Discussion; the sentence on mutation intolerance has also been altered

Figure 1

Enlargement isn't as indicated by the dotted box, and has a different orientation. I had difficulty distinguishing the blues from each other – so could this be clear in the final paper?

Response: The enlargement is actually meant to be rotated by about 90°, as indicated by the circular arrow. This is necessary to show all the details. We have now improved the colours in Fig. 1 and hope it is more clear. Even one of our colour-blind coauthors was happy about it...

Extended data

I couldn't find any methods concerning the AGO2 antibody used in Extended Data Figure 5, nor why it is called alpha AGO2.

Response: The species and origin of the antibody has now been included.

In Extended Data Figure 10, eyeballing the SHANK3 expression data versus the bar graph of SHANK3 density does not seem to correlate – could this be clarified, explained or fixed?

Response: These problems were caused by not all dendrite fragments being shown at the same magnification. We have fixed this problem now, by putting approximately the same size of dendrite fragment for each mutation into the Figure (now: Supplementary Fig. 12). We have removed the overview pictures, as they are already shown in the main Figure (now Fig. 5).

Reviewer #3 (Remarks to the Author):

In this manuscript, Lessel et al. demonstrated the mutations in germline AGO2 affects the function of RISC, in particular for the neurological development by identifying 13 mutations in 20 patients with neurological development diseases. The authors used multi-omics level analysis including quantitative mass spectrometry, which this review will focus on.

Response: We thank the reviewer for their interest and their thoughtful comments and feedback. Below are our responses to the questions and concerns raised by the reviewer.

The quantitative mass spectrometry was used in targeted fashion to measure the phosphorylation level of Ser387 and C-term cluster upon immunoprecipitation of AGO2. The authors argue (with the results shown Fig 3a) that the phosphorylation of C-term cluster was reduced whereas Ser387 phosphorylation is not changed significantly except for H203Q mutation.

However, this analysis is not persuasive as it is unless more information from two major sources are provided.

First, I failed to find the mass spectrometry related method detail to generate the results in Fig 3. The authors are referring to the previous paper (Phosphorylation of Argonaute proteins affects mRNA binding and is essential for microRNA-guided gene silencing in vivo) for the method detail, but the referring paper seems not to contain the MS instrumental set-ups, parameters, or protocols. Did the authors use normal shotgun proteomics or targeted method such as SRM or MRM? Even if this information was provided (and I failed to find) in the previous paper, it should be again provided in this manuscript even briefly.

Response: We completely agree with this reviewer that since methods details (including instrumental set-ups, parameters, and protocols) in the previous paper are within supplementary data, this makes it rather difficult to follow. Thus, for clarity we have now included a complete section on all details of the mass spectroscopy quantification method (SRM).

Second, how the ratio between p vs np was measured is not clear. Did the authors run automated tools or was it done manually? How the phospho peptides were identified? Such information could be useful for readers to understand the results and will make this manuscript more convincing.

Response: This information has now been included in the Methods section.

Minor points for Fig 3a:

1) The y-axis reads "ratio p/n.p. peptide (%)". In legends, please specify p/n.p means phospho vs. non-phospho. Also please explain why the unit is %. The ratio should have no unit in general.

Response: This has now been explained in the Legends as follows:

"The y-axis represents the percentage of individual phosphorylated peptide species assuming the sum of singly, multiply and non-phosphorylated peptides to be 100%"

2) The meaning of whisker in bar plot is not shown. In case of H203Q, the whisker is rather wide; it is not clear this ratio bar has statistical meaning. Please calculate p-value for each case and mark the statistically meaningful ones with *, for example. Provide how the p-values are calculated.

Response: p-values/* marks have now been included where appropriate. Indeed, for the H203Q mutant experimental error was too big to determine any statistically significant difference.

3) The axis font is too small.

Response: This has been improved.

4) Please use the term quantitative mass spectrometry instead of mass spectroscopy.

Response: We thank this reviewer for careful evaluation of our manuscript. This has been changed accordingly.

Reviewer #4 (Remarks to the Author):

In their manuscript "Germline AGO2 mutations impair RNA interference and human neurological development", Lessel et al revealed 13 heterozygous mutations in Ago2 gene in 20 patients with neurological disorders. Although these are heterozygous and do not affect known Ago2 functional domains or modification sites, they affect conserved residues, many are somewhat recurrent (in the sense that 5 of them occurred as independent de novo events), and several are in silico predicted as pathogenic.

Based on this, they conduct assays of the variant Ago2 molecules, primarily by expressing these in Ago2^{-/-} cells or in wildtype primary cells. From these tests, they provide evidence that these mutations can affect the function of Ago2 in different levels. For example, they concluded from their assays that individual single amino acid mutations can impair shRNA-mediated silencing, reduce phosphorylation that may be important for target release, or increase formation of dendritic P-bodies.

They also analyzed transcriptome changes in fibroblasts from several Ago2 patients, suggesting some overlaps although it is hard to draw clear direct conclusions to miRNA functional changes from these tests.

Overall, the clinical significance and dysfunction mechanism for Ago2-mutations revealed by this manuscript make the topic appropriate for Nature Communications. I have the following major concern and a few minor concerns. If they could be addressed, the paper would be a good candidate for publication, but it is somewhat needing further evidence to support the functional conclusion.

Response: We thank the reviewer for their interest and their thoughtful comments and feedback. Below are our responses to the questions and concerns raised by the reviewer.

Major concern.

It is recognized from mouse studies that ago2 is lethal gene, so we don't expect human homozygous patients. However, I am not aware of reported heterozygous ago2 mouse phenotypes. It is certainly possible the human defects require a sensitive phenotypic setting like the brain/behavior to be revealed. However, the model would have to be for the Ago2 mutant proteins to be directly causal, they need be sufficiently dominant negative to cause a problem in the presence of wildtype Ago2 allele.

Response: A dominant negative effect would actually mean that the mutant allele impairs the activity of the WT allele. This is certainly not the case for the G733R mutant, which is a clear loss of function

mutation, and for the large chromosomal deletion observed in patient 21. Furthermore, we have performed additional analyses here to experimentally respond to this issue, by co-expressing mutant AGO2 with WT AGO2 (Fig. 2F). Also here, the analyzed mutations do not exert a dominant negative effect. Actually, this is in line with the finding that most of the *de novo* missense mutations causing similar human disorders result in a loss-of-function rather than in a dominant negative effect.

It would be best if there were knock-in alleles, but I realize that would require some efforts. Alternatively, the existing data needed to be more robust. Instead, their tests are in the setting of misexpression mutant Ago2 proteins, sometimes in an ago2-knockout setting. With these caveat, they need more rigorous tests to (1) showing altered activities, and (2) if plausible to have effects in the physiological setting.

Response: This comment alludes to potential overexpression artefacts in our experiments. To address this we have performed the “titration experiment” which the reviewer suggested (see below), which clearly shows that altered activities of AGO2 WT vs mutants is observed at physiological levels of the protein. Furthermore, we wish to note here that also in the functional experiments in neurons (Fig. 5) we used an expression vector which in our hands induces a rather moderate expression (ubiquitin promoter instead of CMV-promoter). In experiments in fibroblasts, we rely on the endogenous expression levels of AGO2, as no transfections are performed.

For example, they concluded all variant Ago2 proteins show decreased shRNA silencing, but they lack several important controls. There can be different amount of transfection efficiency of the constructs, different effective amount of cell lysates loaded, etc. At the very least, they need to redo the experiments with a co-transfected control construct to assess if there is change to the stability of Ago2 proteins or just variable transfections, and to blot another endogenous protein as a loading control for the knockdown efficiency. Also, they should test some kind of miRNA responsive reporter, since the major kind of Ago2 regulation is miRNAs and not target cleavage.

Response: We have addressed this in several ways; first, for the Shank3 knockdown in Fig. 2A, we have now included the signal from the coexpressed GFP (which is expressed from the shRNA vector) as a loading control. For 2B (DDX1) and 2C (catenin) we have also included tubulin blots to ensure equal loading. Finally for 2C we have also included a blot for cotransfected mRFP. Furthermore, in all of these experiments we used fluorescence microscopy before cell lysis to make sure that >70 % of cells were transfected. This is now mentioned in legends to Figure 2 and S6. Finally, we have now included a miRNA responsive reporter experiment in HeLa cells in the new Supplementary Fig. 10.

Second, they should do some titration experiment in wildtype cells and compare with blotting of Ago2 (to check total Ago2 proteins) that they are not in a big misexpression range. This can evaluate whether they can still see any functional differences in wildtype vs mutant Ago2. This is useful to know if any effect can be seen in a range that is more like the heterozygous state. For this purpose they can just check a couple of the more promising variants they think are more likely to show a functional readout, doesn't need to be all of them. I am trying to provide a reasonable path they could try to show such mutations can be physiologically relevant, even though they don't have a knockin cell model.

Response: We appreciate this comment, as it forced us to more carefully calibrate our experimental system, with, as we believe, quite interesting results. Thus we used the very robust knockdown of GFP- δ -catenin, and took those mutants which had the most severe effect here (F182del; L192P; and G733R). We used three different amounts of AGO2 plasmid for transfection, while keeping the amounts of the other plasmids and the total amount of DNA for each transfection constant. We transfected these into the AGO2 deficient cells, but also used wt 293T cells with the endogenous AGO2 in parallel. These experiments showed that at the upper range we had an about 10- to 20 fold overexpression of AGO2; using the lower amount of plasmid we had about half or less of the amount of AGO2 compared to wt 293T cells. The results of this experiment, as presented in the new supplementary Figure 6, show that (1) the efficiency of knockdown is over a wide range of concentrations independent of the amount of Ago2; and (2) the relation between wt and mutants remain unchanged, meaning that mutants are always lower in silencing efficiency than the wild type, regardless of the amount of AGO2 expressed. This clearly shows that the mutations found here have an effect at physiologically relevant concentrations.

Minor concerns.

1. I am not very familiar with ExAC and gnomAD databases. I thought its very striking that Ago2 is "one of the most missense-intolerant genes in the human genome" 15th/18000". Indeed, ExAC lists Ago2 at 15th. However, it seems ExAC has been replaced/incorporated into the current gnomAD database, which seems to include more data, and states this is the database to use moving forward. In gnomAD, now Ago2 ranks 818 out of 19703 genes. Can the authors comment on the differences of the databases, and why the older ExAC ranking would be appropriate to cite?

Response: The reason why we have referred to the ExAC database, which was not replaced by gnomAD at the time of submission, is the fact that the gene intolerance according to ExAC data has been published, see suppl. table 13 in M. Lek et al., Analysis of protein-coding genetic variation in 60,706 humans. Nature 536, 285-291 (2016), (according to this data AGO2 ranks 15). The manuscript describing the gnomAD data is still "only" in bioRxiv, <https://www.biorxiv.org/content/10.1101/531210v3>, and there is not a corresponding table there to which we can refer.

1. In Figure 1B, the authors identified a patient-related deletion of N-terminal of Ago2 due to a 235.5 kb genomic deletion, but this deletion also removed 23 exons of the C-terminal of PTK2 gene which is upstream of Ago2 gene. Does this deletion (if they are in-frame) cause a gene-fusion of PTK2-Ago2 gene, considering many gene fusions can lead to disease? This should be discussed.

Response: The identified deletion affects the first three AGO2 exons, thus as the start codon is lost, no functional AGO2 protein can be built, resulting in haploinsufficiency. There is indeed a possibility that the deletion may result in gene fusion, which actually might even result in a stable protein. However, there is no way that such putative gene-fused protein would be able to perform similar function to the AGO2 and thus compensate.

Nevertheless, we wanted to tackle this question by performing a RT-qPCR to look for gene-fusion RNA. Unfortunately, no sample was available for this analysis.

2. The y-axis labeling in Figure 4D should be "Ago2 clusters/ μ m dendrite".

Response: This has been changed; this is now Figure 5D.

3. The Western blotting in Supplementary Figure 8 should be replaced by the full image, but not the cropped one.

Response: We have now added the full Western Blot at the bottom of the autoradiography Figure; this is now supplementary Figure 9.

4. This is not really a full written paper, it doesn't have a proper introduction and the paper ends very abruptly without discussion or putting into context the relation of these mutations to other proposed disease-associated mutations in Ago2, and other specific Ago2 amino acids that have been suggested from mechanistic studies to impact Ago2 function.

Response: We have greatly expanded the manuscript and prepared a full length one, including Introduction and Discussion sections. Please note that we had initially submitted to another Journal of the Nature family.

REVIEWER COMMENTS

Reviewer #1 (Remarks to the Author):

The authors have addressed all my inquiries. I find the revised manuscript even more fascinating than the original and congratulate the authors again.

Reviewer #2 (Remarks to the Author):

The authors have satisfactorily addressed the matters I raised.

Reviewer #3 (Remarks to the Author):

All the issues have been well resolved. I recommend to publish this paper in Nature communications journal.

Reviewer #4 (Remarks to the Author):

I thank the authors for thoughtful efforts to improve the robustness of their experimental data to support loss-of-function effects of human Ago2 alleles found in neurological disease. I think these findings will be of interest to the community and I support publication.

I request that the authors revise their attribution of ExAc and gnomAD, which I do not find to be satisfactory. As can be seen, the ExAc database has been retired (<http://exac.broadinstitute.org/>) and is no longer the point of reference ("The ExAC browser is no longer available".) It has been superseded by gnomAD (<https://gnomad.broadinstitute.org/>). As mentioned, Ago2 drops from 15th to 818th in the ranking. bioRxiv publications are a perfectly valid citation to reference (!!), and Broad Institute wants you to use gnomAD, not ExAC. The authors are welcome to mention both the high and low numbers, but not only the high (15th) number, which I feel is misleading, unless there is a statistically sound reason to use the old data, which is based on smaller sample sizes.

Reviewer #5 (Remarks to the Author):

As requested, I only comment on the MD part, which needs more details and information in order to be solid.

Line 243 -> please report the cumulative time of the plain MD simulations

Also, specify how the 3d-model was build (for example, reporting the pdb code). I see this info in the SI, but I would mention these in the text. Or at least call the SI, in the main text.

Like 244-5 -> metadyn requires collective variables that should be specified. This is a critical factor for the proper sampling of the phase space.

So, what is the goal of non-biased vs enhanced sampling simulations? this should be clarified, otherwise, in the main text, I do not get what was the use of “extended” non-biased MD simulations, vs matadyn.

Line 248 -> “ in WT-AGO2 the partial unwinding of the RNA duplex is much larger than in p.L192P”. this should be (and can be) quantified. Is this from metadyn (and thus accelerated through a specific CV), or in unbiased MD?

Also, regarding the following sentences, like “we observed an increased helix-7-MID distance also for the p.F181A mutant”, data must be reported. These qualitative comments are not sufficient. MD simulations allow measuring changes in critical distances, and these should be reported.

In addition, it is not clear to me what is the reference value for such distances, and again, this is critical in order to appreciate if the change is meaningful, or not.

Regarding metadynamics sims: usually, these are used to retrieve the free energy of the process under investigation. This information (like also any additional number from metadyne sims) depends critically on the convergence of such simulations. The convergence of metadyn should be reported in the SI. Also, I find naïve to state – as in the supp info – that “The resulting potential of mean force is not presented “. I do not get what is the use of metadyn, decoupled from the free energy analysis. And, again, the fact that a certain distance value increases, is the obvious result of the simulation (if an obvious CV is used). The authors need to clarify this critical point on the use of metadyn – otherwise, as it is, data from MD simulations, and the use of this comp approach, appear rather immature.

Reviewer #4 (Remarks to the Author):

I thank the authors for thoughtful efforts to improve the robustness of their experimental data to support loss-of-function effects of human Ago2 alleles found in neurological disease. I think these findings will be of interest to the community and I support publication.

I request that the authors revise their attribution of ExAC and gnomAD, which I do not find to be satisfactory. As can be seen, the ExAC database has been retired (<http://exac.broadinstitute.org/>) and is no longer the point of reference ("The ExAC browser is no longer available".) It has been superseded by gnomAD (<https://gnomad.broadinstitute.org/>). As mentioned, Ago2 drops from 15th to 818th in the ranking. bioRxiv publications are a perfectly valid citation to reference (!!), and Broad Institute wants you to use gnomAD, not ExAC. The authors are welcome to mention both the high and low numbers, but not only the high (15th) number, which I feel is misleading, unless there is a statistically sound reason to use the old data, which is based on smaller sample sizes.

Response:

We thank the reviewer for their interest and are pleased that by performing several experiments we have now gained his support for the publication.

Regarding the last remaining issue, and as we have also communicated previously, the main reason for initially referring to ExAC flagship manuscript was the fact that the constraint metrics were included (suppl. table 13 in M. Lek et al., Analysis of protein-coding genetic variation in 60,706 humans. Nature 536, 285-291 (2016). Such supplementary data were NOT available for the gnomAD dataset manuscript that was deposited in bioRxiv. However, as the flagship gnomAD manuscript has now been published, again in Nature, we were finally able to access these data, see supplementary_dataset_11_full_constraint_metrics.tsv, according to which *AGO2* with a missense Z-score of 6.058 ranks 30! This information has now been added in the manuscript.

Reviewer #5 (Remarks to the Author):

As requested, I only comment on the MD part, which needs more details and information in order to be solid.

We thank the reviewer for his thoughtful comments and suggestions. We have now included several new experiments (summarized in the new Figure 4 in the main text, Supplementary Fig. 12-16 and Supplementary video 1) along with methodological details (Methods section in the main text and in the Supplementary Methods).

1. Line 243 -> please report the cumulative time of the plain MD simulations Also, specify how the 3d-model was build (for example, reporting the pdb code). I see this info in the SI, but I would mention these in the text. Or at least call the SI, in the main text.

The minimal trajectory length in this set is 200 ns, and the cumulative simulation time is appr. 17 μ s. This information is now presented in the Materials and Methods section in the main manuscript text. We also mentioned now in the main text/results section the supplemental Figure S12, which lists the underlying structures and pdb codes of the five states we have used: apo-AGO2, two AGO2

complexes with guide RNAs and two AGO2 complexes with guide-target duplexes, simulated for WT and 11 variants.

2. Like 244-5 -> metadyn requires collective variables that should be specified. This is a critical factor for the proper sampling of the phase space.

All metadynamics simulations have now been moved to the supplemental Figure 14; there the structural view of AGO2 has been updated (now: S14a) to illustrate the collective variables used. We also elaborated on these collective variables in the text.

3. So, what is the goal of non-biased vs enhanced sampling simulations? this should be clarified, otherwise, in the main text, I do not get what was the use of “extended” non-biased MD simulations, vs matadyn.

We appreciate this comment, as it points to the fact that our previous Figure 4 mixed the two different approaches in a way which was somewhat confusing. We have now separated non-biased MD simulations, which appear in Figure 4 and Supplementary Fig. 12,13 and 16, and metadyn simulations, which have been moved to Supplementary Fig. 14 and 15.

Our approach, and our goals were as follows: The goal of the non-biased simulations was of course to see what kind of effects the patient-derived mutations have on apo-AGO2 and AGO2-RNA complexes. So we did not focus on any particular process or property of the simulated models. We introduced mutations, then obtained trajectories and analyzed them in different ways, as indicated in the text, and depicted in corresponding figures. We certainly believe that these are “extended” analyses, as we simulated wt and all mutations for five different states of AGO2. Of course, we were interested in trajectories' features that are different between patient-derived mutations and WT.

We saw only few mutations affecting the global protein dynamics (presented in Supplemental Fig 13). Further, we also saw several mutations having two important effects: a destabilizing effect on the helix7-g7 interaction, similar to the mutant reported in 2017 by Klum and MacRae (current ref. 24), and the destabilization of g21-PAZ interaction, supportive for the defect in anchoring of the guide 3'-end in the PAZ domain reported by Jung et al. 2013 (current ref. 33). This entirely new data set has now been included as Figs. 4a+b. We also include the relevant parameter (the I365 α -g7 distance) from ref. 24 for comparison in Fig. 4a.

These observations suggest unwinding of RNA duplexes as a major problem of patient-derived AGO2 mutations. They therefore formed the basis for further investigations using enhanced sampling simulations, where we focused on the unwinding process for wt AGO2 and one of the recurrent patient-derived mutants, p.L192P. We have now tried to make this sequential approach more clear during the course of the MD results section, by clearly separating non-biased and metadynamics parts.

4. Line 248 -> “ in WT-AGO2 the partial unwinding of the RNA duplex is much larger than in p.L192P”. this should be (and can be) quantified. Is this from metadyn (and thus accelerated through a specific CV), or in unbiased MD?

We have now made clear that the partial unwinding was analyzed by the metadynamics approach. As noted in the response to #3, metadyn data are now summarized in its own Figure S14. We added quantitative information on the changes observed in metadynamics simulations to the main MD results section: 13 Å vs. 11.5 Å.

Also, regarding the following sentences, like “we observed an increased helix-7-MID distance also for the p.F181A mutant”, data must be reported. These qualitative comments are not sufficient. MD

simulations allow measuring changes in critical distances, and these should be reported. In addition, it is not clear to me what is the reference value for such distances, and again, this is critical in order to appreciate if the change is meaningful, or not.

We thank the reviewer for this valuable comment. Indeed, in order to clarify the normalized population distributions of the non-biased MD trajectories on $\alpha 7$ -MID we have now included two novel figures: (i) variants bound to fully matched seed region (g2-7 holo-RISC) shown in novel Fig. 4c and (ii) variants bound to a mismatched duplex (g2-8 holo-RISC) shown in the novel Supplementary Fig.16.

Both of these two novel figures show, for each of the analyzed mutations, the calculated critical distances. Both as histograms (always shown in the right panel) with colored circles depicting the maximum population density of each trajectory. Note, that the black dot denotes the maximum population density of the WT AGO2 which is the reference value here. In addition we have included a black cross in this Figure which denotes WT guide-bound complex as a second point of reference. This makes it more clear that the WT duplex-bound complex approaches the WT-guide complex closer, compared to mutations. Please also note that the data are also shown as population histograms on $\alpha 7$ -MID corresponding to the maxima-always on the left panel.

5. Regarding metadynamics sims: usually, these are used to retrieve the free energy of the process under investigation. This information (like also any additional number from metadyn sims) depends critically on the convergence of such simulations. The convergence of metadyn should be reported in the SI. Also, I find naïve to state – as in the supp info – that “The resulting potential of mean force is not presented “. I do not get what is the use of metadyn, decoupled from the free energy analysis. And, again, the fact that a certain distance value increases, is the obvious result of the simulation (if an obvious CV is used). The authors need to clarify this critical point on the use of metadyn – otherwise, as it is, data from MD simulations, and the use of this comp approach, appear rather immature.

We agree with the reviewer that one should always aim at getting converged PMF data from enhanced sampling methods. Unfortunately, it is not always possible, and a reason for this often is not how such methods are performed, but in how complex a system and/or a process is. We were interested in AGO2-mediated unwinding - a process that presumably involves changes on many scales of the protein-RNA complex (from I365 intercalation to global protein dynamics). We believe that performing such task rigorously would be extremely challenging for AGO-RNA complexes, given that free energy calculations of RNA unwinding are not yet straightforward even for isolated RNA duplexes, though this problem is very interesting from both fundamental and practical viewpoints.

During our MetD simulations, the system sampled I365 δ -g(6,7) values partly outside the grid for deposition of the Gaussian potentials because the sampling of the collective variable was not restricted. Thus, the PMF could not be used from this set of simulations. We have now made clear in the SI, and also in the main text, that we could not obtain PMF data for wt and p.L192P mutant AGO2, due to lack of convergence.

However, enhanced sampling can still be useful even without calculating PMF, as trajectories from not fully converged MetD simulations (as in our case) can be analyzed to reveal mechanistic properties of the process induced in the MetD simulations (S. Haldar et al, J. Chem. Theory Comput, DOI: 10.1021/acs.jctc.5b00010). This is rather an unusual application of MetD, as also noted by the Reviewer. Nevertheless, these data are still reliable, clearly under restrictive qualitative conclusion. Thus, enabling us to determine e.g. how changes of some properties during the process are connected to some other properties (connection of helix7 and unwinding). Namely, we have used trajectories from this dataset only to define a measure of unwinding, actually the $\langle C1'-C1' \rangle$

distance, as a function of helix7-MID distance – two variables that were not biased in these simulations. We have included a clear description of this twist in the revised manuscript.

REVIEWER COMMENTS

Reviewer #5 (Remarks to the Author):

The authors have received my comments, and suggestions.

However, I still see that MD simulations and the MD-based data are presented and interpreted in a quite naive manner. I have issues with this use of MD, which is a powerful and quantitative method.

For example:

In the section “Germline AGO2 mutations alter AGO2 unwinding function.”, the unbiased MD simulations are presented. Results are reported and I note that the entire paragraph does not contain a single quantitative estimation in support of the claims. Data are all reported in the supplementary information, and it’s not immediate to see how data support the claims. If the manuscript stays like this, it should be made very clear that these indications are highly qualitative (thus speculative), although somehow in line with the experimental data.

In metadyn: “In WT-AGO2 the highest duplex width reached is larger than in p.L192P (13 Å vs. 11.5 Å, respectively),...”

Numbers from MD should be associated with their deviations, and given the minimal difference between 11.5 and 13, the authors should prove that the delta is okay in consideration of the associated deviation in MD. This is basic in the proper use of MD data.

“Furthermore, we observed a larger helix-7-MID distance also for the p.F181A positive control, but not for the common non-synonymous AGO2 variant p.E186K. “

Again, “larger” should be defined and quantified. Maybe report the “larger” distance in parenthesis.

I may argue on the use of metadyn, again. There is really little one can get from these simulations, more than just a qualitative indication of what may be the process if biased simulations are not converged or extensively repeated, at least. This comment apart, my main point is that MD simulations should be used to retrieve quantitative estimates of the event under investigation. Instead, I can’t see a single number in the entire result paragraph on MD. I get that MD simulations is not the core scientific piece of this story. Yet, I have problems with these MD-based data used in this manner, without a clear statement of the qualitative nature of such MD-based investigations.

Given the enthusiasm for this paper from the other reviewers, I am not against it, of course. I just ask the authors to use and present MD data in a fair and more scientific manner, and at least state very clearly that in the context of this multidisciplinary study, results from MD are highly qualitative, and thus highly speculative.

I am a strong supporter of MD, since this is the tool I use the most in my research. For this very reason, I have issues with such a qualitative use of this powerful method.

Reviewer #5 (Remarks to the Author):

The authors have received my comments, and suggestions. However, I still see that MD simulations and the MD-based data are presented and interpreted in a quite naive manner. I have issues with this use of MD, which is a powerful and quantitative method.

For example:

In the section “Germline AGO2 mutations alter AGO2 unwinding function.”, the unbiased MD simulations are presented. Results are reported and I note that the entire paragraph does not contain a single quantitative estimation in support of the claims. Data are all reported in the supplementary information, and it’s not immediate to see how data support the claims. If the manuscript stays like this, it should be made very clear that these indications are highly qualitative (thus speculative), although somehow in line with the experimental data.

Response: We thank the reviewer for helping us clarify our simulation data. Following his suggestion we have now at several positions in the Results section softened our wording, to make clear that the MD data are qualitative (and therefore somewhat speculative). In addition we have added at the end of this paragraph the sentence: “However, one should keep in mind that our data are mostly qualitative, and further, more quantitative simulations will be needed to completely dissect how the mutations in AGO2 affect unwinding”.

It is worth noting that we performed here a thorough screen of altogether 11 mutations and the WT in 5 different states in order to gain a further insight into the pathomechanism of the mutations identified in patients. We clearly agree with this reviewer that such approach is inherently qualitative, as there is no hope to obtain multiple long replicas for all 60+ trajectories. The computational cost of it would exceed all reasonable amounts. We have briefly mentioned this strategy at the beginning of the MD section.

In metadyn: “In WT-AGO2 the highest duplex width reached is larger than in p.L192P (13 Å vs. 11.5 Å, respectively),...”

Numbers from MD should be associated with their deviations, and given the minimal difference between 11.5 and 13, the authors should prove that the delta is okay in consideration of the associated deviation in MD. This is basic in the proper use of MD data.

Response: In line with the other criticism by this reviewer, we agree completely that our data (non-biased as well as metadynamics simulations) are qualitative rather than quantitative. We think that standard deviations cannot be provided based on qualitative data, as they would convey a false sense of accuracy which is not justified by the nature of our simulations.

“Furthermore, we observed a larger helix-7-MID distance also for the p.F181A positive control, but not for the common non-synonymous AGO2 variant p.E186K. “

Again, “larger” should be defined and quantified. Maybe report the “larger” distance in parenthesis.

Response: we have included this increase in distance here in parentheses, as suggested by the reviewer.

I may argue on the use of metadyn, again. There is really little one can get from these simulations, more than just a qualitative indication of what may be the process if biased simulations are not converged or extensively repeated, at least. This comment apart, my main point is that MD simulations should be used to retrieve quantitative estimates of the event under investigation. Instead, I can’t see a single number in the entire result paragraph on MD. I get that MD simulations is not the core scientific piece of this story. Yet, I have problems with these MD-based data used in this manner, without a clear statement of the qualitative nature of such MD-based investigations.

Response: We have now made it clear at several positions along the MD paragraph that our data are qualitative. As such, we have also refrained from using standard deviations, as we did not perform enough repeats to do so. Again, we would like to mention here that our MD simulations should be seen as a first qualitative approach to an understanding of the functional effects of the mutations in AGO2 we have found in patients.

Given the enthusiasm for this paper from the other reviewers, I am not against it, of course. I just ask the authors to use and present MD data in a fair and more scientific manner, and at least state very clearly that in the context of this multidisciplinary study, results from MD are highly qualitative, and thus highly speculative.

Response: We thank the reviewer for this encouraging words. As already stated above we followed the reviewer's suggestion and clearly stated the MD results to be speculative at this point.

I am a strong supporter of MD, since this is the tool I use the most in my research. For this very reason, I have issues with such a qualitative use of this powerful method.

REVIEWERS' COMMENTS:

Reviewer #5 (Remarks to the Author):

I thank the authors for having addressed most of my concerns.
However, I have one last minor but crucial modification to ask/suggest.

The authors state that they do not feel appropriate to report the error associated to the difference of 13 vs 11.5, from MD.

I am okay if they prefer not to report the standard deviation, however the sentence in the paper that now reads as it follows:

"In WT-AGO2 the highest duplex width reached is larger than in p.L192P (13 Å vs. 11.5 Å, respectively),.."

should therefore become:

"In WT-AGO2 the highest duplex width reached is slightly larger than in p.L192P (~13 Å vs. ~11.5 Å, respectively),.."

I added the word "slightly" and added the ~ symbol, to indicate that numbers are indicative.

The authors may of course prefer a different wording, and I have no prob with that as long as they recognize the limits of this quantification. 1.5 Å is really a minor difference, as I am sure the authors realized already.

	EDITORIAL REQUESTS:	AUTHOR RESPONSE:
1.	Data presentation: Please ensure that data presented in a plot, chart or other visual representation format shows data distribution clearly (e.g. dot plots, box-and-whisker plots). When using bar charts, please overlay the corresponding data points (as dot plots) whenever possible and always for $n \leq 10$. (Please see the following editorial for the rationale behind this request and an example https://www.nature.com/articles/s41551-017-0079).	
	Panels requiring revision:  Please note that data presentation has to be revised to comply with our policy in figures 2d, e; 3b; and supplementary figures 4; 10. 	We have now included dot plots for 2 d, e, 3 a,b, and Supplementary Figures 4 and 10.
2.	Statistics: Wherever statistics have been derived (e.g. error bars, box plots, statistical significance) the legend needs to provide and define the n number (i.e. the sample size used to derive statistics) as a precise value (not a range), using the wording “n=X biologically independent samples/animals/cells/independent experiments/n= X cells examined over Y independent experiments” etc. as applicable.	
	Legends requiring revision:  Please note that this information is missing in the legends of figures 2e, f; 3a, b. Although ‘n’ is provided, please describe the nature of entity for ‘n’ in the figures 2d; 5c, d, e; and supplementary figures 6; 17. 	n-number has now been included, and has been defined in all Figures mentioned here.
3.	Statistics such as error bars, significance and p values cannot be derived from $n < 3$ and must be removed from all such cases.	
	We strongly discourage deriving statistics from technical replicates, unless there is a clear scientific justification for why providing this information is important. Conflating technical and biological variability, e.g., by pooling technically replicates samples across independent experiments is strongly discouraged. (For examples of expected description of statistics in figure legends, please see the following https://www.nature.com/articles/s41467-019-11636-5 or https://www.nature.com/articles/s41467-019-11510-4).	
	All error bars need to be defined in the legends (e.g. SD, SEM) together with a measure of centre (e.g. mean, median). For example, the legends should state something along the lines of “Data are presented as mean values +/- SEM” as appropriate. All box plots need to be defined in the legends in terms of minima, maxima, centre, bounds of box and whiskers and percentile.	
	Legends requiring revision:  Please note that the error bars need to be defined in the legends of figures 2e, f; 3a, b; and supplementary figures 4; 6; 10. 	Error bars have been defined in 2e, f, 3a, 3b, and in Supplementary Figures 4, 6, 10
4.	The figure legends must indicate the statistical test used. Where appropriate, please indicate in the figure legends whether the statistical tests were one-sided or two-sided and whether adjustments were made for multiple comparisons.	

	For null hypothesis testing, please indicate the test statistic (e.g. F, t, r) with confidence intervals, effect sizes, degrees of freedom and P values noted. Please provide the test results (e.g. P values) as exact values whenever possible and with confidence intervals noted.	
	Legends requiring revision:  1. Please indicate the statistical test used for data analysis and where appropriate, please specify whether it was one-sided or two-sided and whether adjustments were made for multiple comparisons, in the legends of figures 2e, f; and supplementary tables 3; 5. 2. Please note that the information on whether the statistical test used was one-sided or two-sided, where appropriate, is missing in the legends of figures 2d; 5c, d, e. 3. Please note that the exact p value should be provided, when possible, in the legends of figures 2d; 3a, b; 5c, d, e; and supplementary figure 10. 	 1. For 2e, f, and supplementary tables 3 and 5 the details of the statistical tests have been indicated. One-sided/two-sided does not apply to ANOVA, as used in 2e, f. 2. also in 2d, 5c, d, e ANOVA was used, therefore this is not appropriate here. 3. exact p-values have now been added to all these Figures within the Figure itself.
5.	Reproducibility: Please state in the legends how many times each experiment was repeated independently with similar results. This is needed for all experiments, but is particularly important wherever results from representative experiments (such as micrographs) are shown. If space in the legends is limiting, this information can be included in a section titled "Statistics and Reproducibility" in the methods section. Legends requiring revision:  1. Please note that this information is missing in the figure legends of 2a, b, c; 5a, b; and supplementary figures 7; 8; 9; 11. 	This information has now been added to Figs. 2a-c, 5a, b; and Supplementary Figure 7, 8, 9 and 11.
6.	Data availability: This journal strongly supports public availability of data and custom code associated with the paper in a persistent repository where they can be freely and enduringly accessed or as a supplementary data file when no appropriate repository is available. If data and code can only be shared on request, please explain why in your data Availability Statement, and also in the correspondence with your editor. For more information, please refer to https://www.nature.com/nature-research/editorial-policies/reporting-standards#availability-of-data Please ensure that datasets deposited in public repositories are now publicly accessible, and that accession codes or DOI are provided in the "Data Availability" section. As long as these datasets are not public, we cannot proceed with the acceptance of your paper. For data that have been obtained from publicly available sources, please provide a URL and the specific data product name in the data availability statement. Data with a DOI should be further cited in the methods reference section.	Datasets in public repositories (RNA seq data, and mass spectroscopy/proteomics data are now publicly available under the sources indicated.
7.	Gels and Blots: Quantitative comparisons between samples on different gels/blots are discouraged; if this is unavoidable, the figure legend must state that the samples derive from the same experiment and that gels/blots were processed in parallel.	

	Vertically sliced images that juxtapose lanes that were non-adjacent in the gel must have a clear separation or a black line delineating the boundary between the gels. Loading controls (e.g. GAPDH, actin) must be run on the same blot. Sample processing controls run on different gels must be identified as such in the figure legends, and distinctly from loading controls. All blots and gels must be accompanied by the locations of molecular weight/size markers. Blots should be cropped such that at least one marker position is present. Please also supply uncropped and unprocessed scans of the most important blots in the Source Data file or as a supplementary figure in the Supplementary Information. This should be cited once in the Methods section. For an example of presentation of full scan blots, see the Source Data file of https://www.nature.com/articles/s41467-020-16984-1#Sec35 and for more information, please refer to https://www.nature.com/nature-research/editorial-policies/image-integrity	
	Panels requiring revision:  1. Molecular weight markers are missing for panels 2a, b, c; and supplementary figures 6; 7; 8. 2. Units of molecular weight markers are missing for supplementary figure 5. 	Molecular weight markers have been included, and the units (kDa) have also been added.
8.	Micrographs: Please ensure that all micrographs include a scale bar and this scale bar is defined on the panels or in the figure legends. Panels requiring revision:  1. Please note that the scale bar needs to be defined for figures 5a, b; and supplementary figures 11; 17. 	Scale bar definitions have been included for all micrographs.